# Manipulating coordination environment for a high-voltage aqueous copper-chlorine battery

Xiangyong Zhang[1,2], Hua Wei [1,2], Shizhen Li[1,2], Baohui Ren[1,2], Jingjing Jiang[1,2], Guangmeng Qu[2], Haiming Lv[2], Guojin Liang[3], Guangming Chen [1], Chunyi Zhi [2,3] ✉, Hongfei Li [4] ✉ & Zhuoxin Liu [1] ✉

Aqueous copper-based batteries have many favourable properties and have thus attracted considerable attention, but their application is limited by their low operating voltage originating from the high potential of copper negative electrode (0.34 V vs. standard hydrogen electrode). Herein, we propose a coordination strategy for reducing the intrinsic negative electrode redox potential in aqueous copper-based batteries and thus improving their operating voltage. This is achieved by establishing an appropriate coordination environment through the electrolyte tailoring via $Cl^-$ ions. When coordinated with chlorine, the intermediate $Cu^+$ ions in aqueous electrolytes are successfully stabilized and the electrochemical process is decoupled into two separate redox reactions involving $Cu^{2+}/Cu^+$ and $Cu^+/Cu^0$; $Cu^+/Cu^0$ results in a redox potential approximately 0.3 V lower than that for $Cu^{2+}/Cu^0$. Compared to the coordination with water, the coordination with chlorine also results in higher copper utilization, more rapid redox kinetics, and superior cycle stability. An aqueous copper-chlorine battery, harnessing $Cl^-/Cl^0$ redox reaction at the positive electrode, is discovered to have a high discharge voltage of 1.3 V, and retains 77.4% of initial capacity after 10,000 cycles. This work may open up an avenue to boosting the voltage and energy of aqueous copper batteries.

The increasing demand for electrical energy is creating an urgent need to develop sustainable, high-performance, and inexpensive electrical energy storage technologies[1–3]. Lithium-ion batteries (LIBs) are the predominant choice of energy storage, but their comprehensive application has been compromised by the world's limited lithium resources, the negative environmental impacts of LIB production, and safety concerns that arise during LIB operation[4,5]. As a promising alternative, aqueous rechargeable batteries have received tremendous attention due to the intrinsic merits of water-based electrolytes, such as their low cost, high safety, considerable ionic conductivity, and high compatibility with the natural environment[6–9].

Aqueous rechargeable batteries in which naturally abundant alkaline ions ($Na^+$ and $K^+$) and multivalent cations ($Mg^{2+}$, $Ca^{2+}$, $Zn^{2+}$, $Al^{3+}$, $Mn^{2+}$, etc.) are employed as charge carriers have been extensively investigated[10–17]. Among the candidate systems, aqueous copper-based batteries are attracting considerable attention owing to the distinctive merits of copper negative electrodes, such as their high theoretical specific capacities (gravimetric capacity of 844 mAh g$^{-1}$ and volumetric capacity of 7558 mAh cm$^{-3}$), favourable stability, and the abundance of

[1]College of Materials Science and Engineering, Shenzhen University, 518055 Shenzhen, China. [2]Songshan Lake Materials Laboratory, 523808 Dongguan, Guangdong, China. [3]Department of Materials Science and Engineering, City University of Hong Kong, 83 Tat Chee Avenue, 999077 Kowloon, Hong Kong, China. [4]School of System Design and Intelligent Manufacturing, Southern University of Science and Technology, 518055 Shenzhen, China. ✉e-mail: cy.zhi@cityu.edu.hk; lihf@sustech.edu.cn; liuzhuoxin@szu.edu.cn

copper in the earth[18,19]. Recently, some work on positive electrode materials for copper-based batteries has been reported. For example, Prussian blue analogue, an open-framework positive electrode, exhibited high cycling stability and ultrafast kinetics due to the intercalation of $Cu^{2+}$ ions[20]. To achieve high energy density, sulphur and selenium with conversion mechanisms were employed as positive electrodes, which enabled the realisation of four-electron transfer based on redox-active $Cu^{2+}$ ions, resulting in high discharge-specific capacity[21,22]. Although some improvements have been made in terms of discharge capacity and power delivery, the relatively high potential of copper negative electrode (-0.34 V vs. the standard hydrogen electrode, SHE) in these copper-based batteries has generally resulted in low full-cell voltage (-1.0 V), substantially limiting the energy output and further practical application of copper-based batteries.

The matching of high-voltage positive electrodes ($MnO_2$, $PbO_2$, etc.) is currently the main strategy being used to increase the operating voltage of copper-based batteries, but excessive positive electrode potential can lead to parasitic reactions such as the oxygen evolution reaction[18,23,24]. During electrochemical processes involving a copper negative electrode, copper ions naturally exist in two valence states ($Cu^+$ and $Cu^{2+}$), but the $Cu^+$ ions are unstable and prone to disproportionation in which $Cu^0$ and $Cu^{2+}$ are formed[25]. Thus, the negative electrode potential of 0.34 V (vs. SHE) is actually based on the reaction between $Cu^{2+}$ and $Cu^0$. However, as revealed in the early studies on copper-based flow batteries, $Cu^+$ ions can be stabilised through coordination chemistry via acetonitrile or halogen, and a lower redox potential for $Cu^+$ and $Cu^0$ can thereby be identified[26–28]. On the basis of this discovery, we propose that rationally manipulating the coordination environment to reduce the intrinsic negative electrode redox potential may be an effective approach to improving the operating voltage of copper-based batteries.

In this study, we initiated an investigation of the electrochemical kinetics and behaviours of copper ions under the coordination of chloride and water molecules. Compared with water coordination, chlorine coordination, i.e., an electrolyte environment that enables the coordination of chloride with copper ions, led to higher copper utilisation, more rapid redox kinetics, and greater cycle stability. More importantly, it resulted in reduced negative electrode potential that suggests the possibility of constructing aqueous copper-based batteries with higher operating voltage. The relatively low negative electrode potential is attributable to the chloride coordination of copper ions; the intermediate $Cu^+$ ions are favourably stabilised, and the electrochemical process is thereby decoupled into two separate redox reactions involving $Cu^{2+}/Cu^+$ and $Cu^+/Cu^0$. An aqueous $Cu-Cl_2$ battery, with the utilisation of $Cl^-/Cl^0$ as redox reaction for the positive electrode, was successfully built and thoroughly investigated. The resultant full cell was discovered to have a high discharge voltage of 1.3 V, which is higher than that of any previously reported aqueous copper-based battery system. Additionally, the cell manifested a superior discharge capacity of 162 mAh $g^{-1}$ at a specific current of 5 A $g^{-1}$, and retained 77.4% of its initial capacity after 10,000 charge–discharge cycles at 20 A $g^{-1}$ with an upper cutoff voltage of 1.6 V.

## Results
### The electrochemical characterisations of Cu negative electrode in different electrolytes
To initially test the proposed hypothesis, we employed various halogen-containing electrolytes to alter the coordination environment of copper ions. Cyclic voltammetry (CV) test results obtained using different halogen electrolytes (4 M NaX and 0.05 M $CuX_2$, where X = Cl, Br, or I) are compared in Fig. 1a. Despite the use of saturated NaF (around 1 M) and $CuF_2$ solution, the solubility of $Cu^{2+}$ was very low; thus, the corresponding CV curve does not contain a distinct redox peak (Supplementary Fig. 1). Because $I^-$ spontaneously reduced $Cu^{2+}$ into $Cu^+$, only the redox reaction of $Cu^+/Cu^0$ was examined (Supplementary Fig. 2). Under the coordination of halide ions ($Cl^-$ and $Br^-$), the CV curve contained two pairs of redox peaks corresponding to the reactions of $Cu^{2+}/Cu^+$ and $Cu^+/Cu^0$, indicating that the formation of halide complexes effectively stabilised the $Cu^+$ ions. More importantly, the stabilised $Cu^+$ exhibited notably low reduction potential, which suggests the possibility of constructing aqueous copper-based batteries with higher operating voltage than already reported. The shift in the oxidation or reduction potential of $Cu^+$ is actually related to the stability constant of the complex that formed under the altered coordination environment (Fig. 1b)[29]; the higher the stability constant, the more difficult it is for the oxidation or reduction reaction to occur, resulting in higher oxidation potential and lower reduction potential. Compared with that in the electrolyte not containing chloride, the redox pair in the chloride-containing electrolyte appeared at approximately 0.3 V lower potential (Fig. 1c), that is, if the $Cu^+/Cu^0$ reaction was activated, the negative electrode potential could be reduced by 0.3 V.

### Copper ion behaviours under different coordination environments
To investigate the effect of the coordination environment on the electrochemical behaviour of copper ions, CV measurements were conducted on carbon cloth electrodes employed in a standard three-electrode cell. The chlorine coordination was achieved using an aqueous solution comprising 4 M NaCl and 0.05 M $CuCl_2$ as the electrolyte (denoted Cu−Cl), whereas the water coordination was achieved using a solution comprising 2 M $Na_2SO_4$ and 0.05 M $CuSO_4$ as the electrolyte (denoted Cu−$H_2O$). The decision to utilise a 4 M NaCl and 0.05 M $CuCl_2$ electrolyte was informed by its resultant high areal capacity, Coulombic efficiency, and good reversibility of the redox reactions (Supplementary Figs. 3 and 4). The nominal potential of all redox couples was estimated by taking the average of the observed reduction and oxidation peak potentials. As illustrated in Fig. 2a, the redox behaviour of copper with water coordination revealed a redox potential at 0.02 V versus the saturated calomel electrode (SCE), which corresponded to the $Cu^{2+}/Cu^0$ reaction. In the Cu−Cl electrolyte, two pairs of redox peaks were discovered due to stabilisation of $Cu^+$ by the chlorine coordination. The redox doublets located at standard potentials of approximately −0.28 V were generated by the conversion of $Cu^+/Cu^0$, whereas the doublets at approximately 0.36 V were related to the $Cu^{2+}/Cu^+$ redox couple. The emergence of this lower-potential redox pair for $Cu^+/Cu^0$ means that it is theoretically possible to construct an aqueous copper battery with boosted full-cell voltage. To further determine the effect of cations on a cell's electrochemical properties, CV tests involving different chloride salts (KCl and $CaCl_2$) were performed, and cation species were discovered to have little effect on the redox behaviour of copper ions (Supplementary Figs. 5 and 6).

As shown in Fig. 2b, surface-enhanced Raman scattering (SERS) spectra revealed distinct coordination patterns between copper ions and the respective electrolytes. In the Cu−$H_2O$ electrolyte, copper ions were primarily observed to coordinate with water molecules[30]. However, in the Cu−Cl electrolyte, the coordination between copper ions and water was significantly weakened, and the predominant coordination was observed with chlorine[19,31]. Notably, Cu−Cl coordination does not necessitate exceedingly high chloride ion concentrations and generally occurs at concentrations above 1 M (Supplementary Fig. 7). This is due to the higher negative charge and larger size of chloride ions, which allow for the formation of stronger bonds with positively charged copper ions compared to water molecules.

The change in coordination patterns was supported by the Nuclear magnetic resonance (NMR) spectra shown in Fig. 2c, where the peak corresponding to the coordination water in the Cu−Cl electrolyte

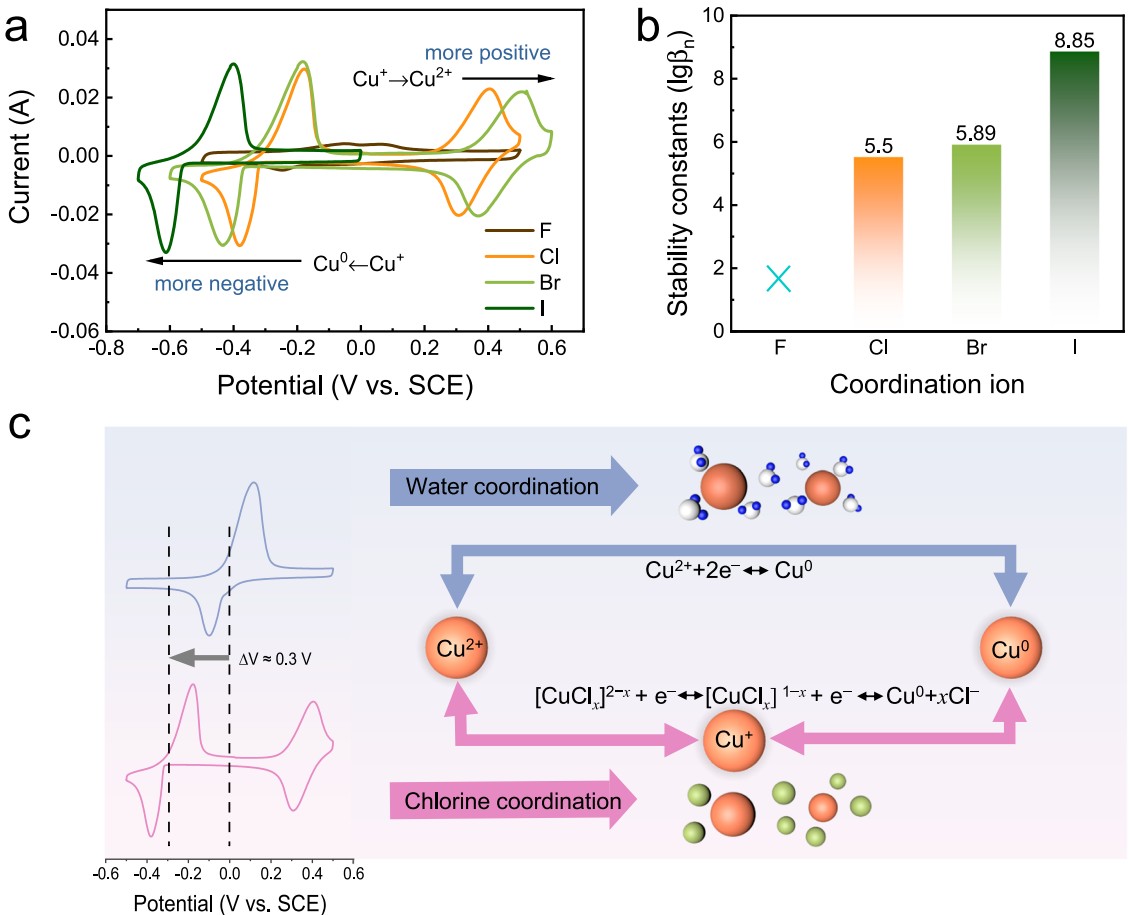

**Fig. 1 | The electrochemical characterisations of Cu negative electrode in different electrolytes. a** CV curves scanned at 10 mV s⁻¹ in various halogen electrolytes. **b** Stability constant of different halogen complexes. **c** Illustration of the differences in redox behaviour under chlorine and water coordination. The electrochemical tests were conducted in a three-electrode cell consisting of a carbon felt working electrode, a Pt plate (1 × 1 cm) counter electrode, and an SCE reference electrode.

exhibited a shift compared to the Cu−H₂O electrolyte, indicating a decrease in coordination water and the partial substitution by chloride ions[32,33]. In addition, the ultraviolet−visible (UV-Vis) spectra were employed to further analyse the coordination patterns of copper ions under different electrolytes. The presence of a peak near 255 nm (Fig. 2d) indicated the formation of Cu−Cl complexes in the Cu−Cl electrolyte, primarily in the form of CuCl⁺ and CuCl₂⁰.[30,34]

To gain a deeper understanding of the coordination chemistry, we conducted molecular dynamics simulations (shown in Fig. 2e, f and Supplementary Fig. 8). The simulated results revealed that in the Cu−H₂O electrolyte, copper ions coordinated with six water molecules. While in the Cu−Cl electrolyte, the copper-water-chloride complex consisted of approximately 1.8 chlorine ions and 4.2 water molecules. This finding suggests that in the CuCl₂ − NaCl solution with 4 M NaCl, a chloride-rich hydration sheath of copper was formed rather than the complete hydration of copper ions.

**Copper ion redox mechanism in Cu−Cl electrolyte**

Ex situ X-ray diffraction (XRD) and UV-Vis spectrophotometry were conducted to gain insight into the electrochemical mechanism underlying the copper redox reactions. Under various potentials over the CV curve (1 mV s⁻¹, Fig. 3a), XRD patterns were obtained and presented in Fig. 3b. At the beginning of the CV negative scan (I → II), no new solid phase formed on the carbon electrode, indicating that the reduction peak corresponded to a liquid−liquid reaction. As the potential approached −0.5 V (vs. SCE), pure copper (PDF#04-0836) appeared (II → III) and then gradually dissolved with the progression of the CV positive scan (III → IV and IV → V), indicating that the redox pair

at lower potential corresponded to a solid−liquid reaction. UV-Vis spectra (Fig. 3c) were also collected at various potentials selected on the basis of the CV test result for a lower concentration of copper ions (1.2 mM, 1 mV s⁻¹, Supplementary Fig. 9). Importantly, the coordination behaviour of Cu in chloride forms is primarily determined by the concentration of free chloride ions rather than the total Cu concentration[35]. Therefore, to stabilise the concentration of free chloride ions and minimise any changes in copper chloride coordination, we diluted the copper ion concentration while maintained a high concentration of chloride ions at 4 M, thus we were able to obtain absorption spectra that provided reliable information about the coordination chemistry and the copper electrochemical behaviour. The initial spectrum (I) had only one peak at ~255 nm, which was related to Cu²⁺ ions[36]. As the CV negative scan reached −0.1 V (II), a new peak corresponding to Cu⁺ emerged at 270 nm[37–39]. In the traces of redox behaviour, Cu⁺ appeared through the first reduction reaction and disappeared through the last oxidation process, indicating that the high-potential redox pair is associated with the Cu²⁺/Cu⁺ reaction. During the lower-potential redox process, the relative intensity of Cu⁺ first decreased and then increased.

Meanwhile, the ex situ X-ray photoelectron spectroscopy (XPS) spectra revealed distinct peaks corresponding to different copper species. As shown in Fig. 3d, the Cu 2p₃/₂ and Cu 2p₁/₂ peaks were observed at approximately 935.2 and 955 eV, respectively, indicating the presence of Cu²⁺ species. While the Cu 2p₃/₂ and Cu 2p₁/₂ peaks at 932.3 and 952 eV, respectively, were attributed to Cu⁺ and/or metallic Cu⁰ species[40,41]. Thus, these peaks detected at II, III, and IV states indicate the formation of Cu⁺ and/or Cu⁰ species during discharging.

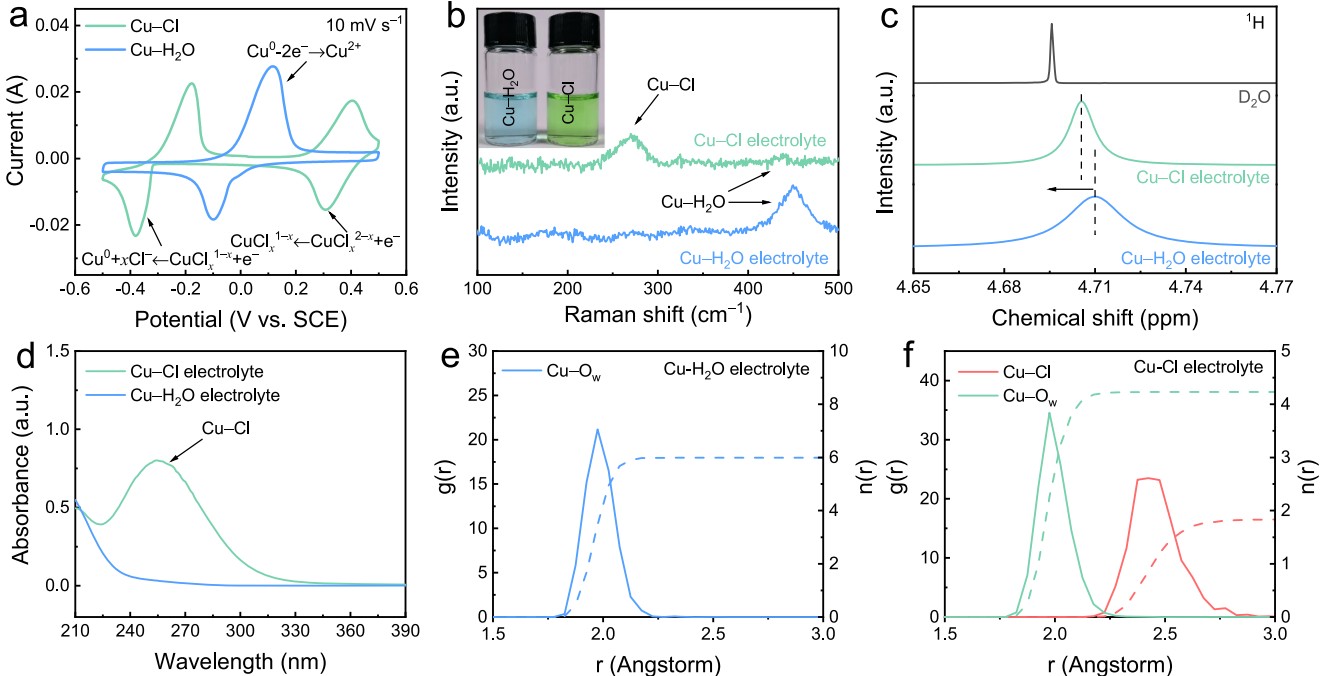

**Fig. 2 | Copper ion behaviours under different coordination environments. a** CV curves obtained at a scan rate of 10 mV s⁻¹, (**b**) SERS spectra, (**c**) ¹H NMR spectra, and (**d**) UV-Vis spectra of Cu–Cl and Cu–H₂O electrolytes. **e** RDF of Cu–H₂O electrolyte. **f** RDF of Cu–Cl electrolyte. (The solid lines are the radial distribution functions, and the dotted lines are the coordination numbers.) The electrochemical tests were conducted in a three-electrode cell consisting of a carbon felt working electrode, a Pt plate (1 × 1 cm) counter electrode, and an SCE reference electrode.

Nevertheless, due to the small difference in binding energies, it is still challenging to distinguish between the chemical states of zero-valent copper (Cu⁰) and univalent copper (Cu⁺) based solely on these XPS spectra[42,43]. To gain further insights, X-ray excited Auger Electron Spectroscopy (XAES) characterisations were conducted. As shown in Fig. 3e, the Cu *LMM* peak fitting analysis reveals distinct peaks corresponding to different copper species. Specifically, the peak centred around 918.5 eV was consistent with metallic Cu⁰, while the peak at 915.5 eV was attributed to Cu⁺. Additionally, the peak at approximately 917 eV was associated with Cu²⁺ ions[44,45]. Supplementary Fig. 10 further demonstrated similar valence changes through direct comparison of XPS and XAES spectra between the electrodes after copper deposition and charging to 0.3 V.

Combining the XRD and UV-Vis analysis, these additional ex situ characterisations provide robust evidence for the reversible redox reaction of the Cu negative electrode: in the initial state (I), the valence state of copper (Cu) is primarily +2. As the CV negative scan proceeds (I → III), the predominant Cu²⁺ species undergoes partial reduction to Cu⁺/Cu⁰. Subsequently, during the subsequent positive scan process (III → V), the Cu⁰ species is oxidised back to Cu⁺ and further to Cu²⁺. These reactions resemble the previously reported transformations involving solid CuCl₂, NaCl, and CuCl[46]. Briefly, the electrochemical processes of copper ions under chlorine coordination can be described as follows ($x = 1 - 4$, which represents the coordination number):

$$[CuCl_x]^{2-x} + e^- \leftrightarrow [CuCl_x]^{1-x} \quad (1)$$

$$[CuCl_x]^{1-x} + e^- \leftrightarrow Cu^0 + xCl^- \quad (2)$$

Furthermore, to substantiate the proposed mechanism, theoretical simulations were conducted to investigate the reaction pathways of Cu–H₂O and Cu–Cl electrolytes. The initial reactants employed in the simulations were determined based on the results of radial distribution function (RDF) calculations. Specifically, the Cu–H₂O electrolyte reactant was modelled as Cu(H₂O)₆²⁺, while the Cu–Cl electrolyte reactant was represented as CuCl₂(H₂O)₄. As shown in Fig. 3f, in the Cu–H₂O electrolyte, the simulations revealed a direct conversion pathway from Cu(H₂O)₆²⁺ to Cu(H₂O)₆, as the proposed intermediate species Cu(H₂O)₆⁺ exhibited a higher energy of 5.1 eV and readily transformed into the more thermodynamically stable Cu(H₂O)₆ (2.6 eV). In contrast, in the Cu–Cl electrolyte, the simulation results uncovered a distinct reaction pathway initiated by the formation of CuCl(H₂O)₄ from CuCl₂(H₂O)₄, involving the surmounting of a 0.9 eV thermodynamic energy barrier. Subsequently, the formed CuCl(H₂O)₄ underwent further reactions, ultimately leading to the formation of Cu(H₂O)₄. Notably, these simulation findings closely aligned with our CV data, offering robust support for the proposed reaction mechanism.

The copper deposition–dissolution behaviour of the negative electrode reaction was subsequently investigated under chlorine coordination and water coordination. In the Cu 2*p* XPS spectrum presented in Supplementary Fig. 11, the peaks located at 932.6 and 952.4 eV represent metallic copper, which further indicates that the deposition product was the same under chlorine coordination versus water coordination: pure copper. The surface morphologies of the copper deposited on the carbon felt in each coordination environment were examined separately after a certain electrochemical process under identical current density in the corresponding electrolytes. For the chlorine coordination (Supplementary Fig. 12a), the deposited metallic copper was relatively uniform and appeared as relatively large particles (~10 μm) without dendrites sticking out. By contrast, the copper deposited under water coordination (Supplementary Fig. 12b) was disorganised, rougher, and had whiskers that stuck out. Such morphological characteristics may account for the difference in cycle stability between these two copper deposits, which is discussed later.

## The electrochemical performance of Cu negative electrode
To better understand the electrochemical kinetic behaviours that occurred in these two coordination environments, CV scans were

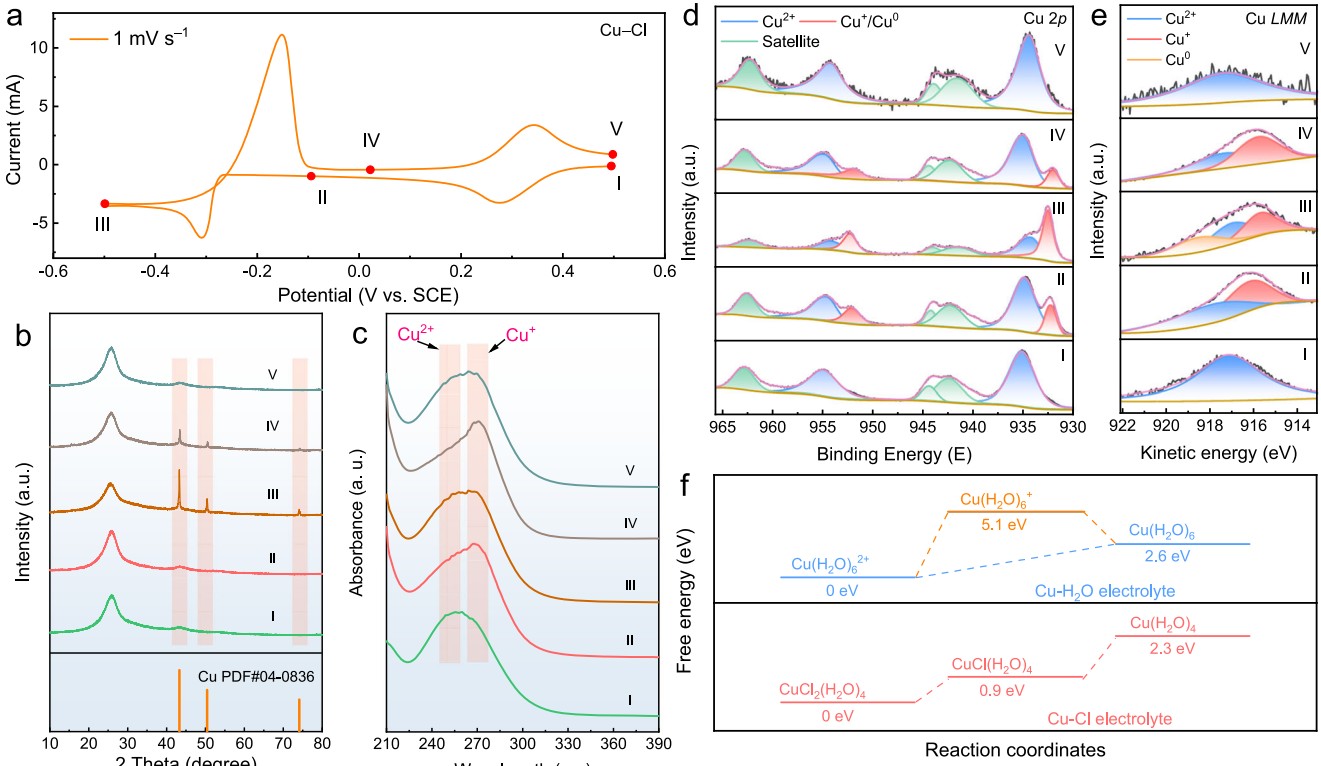

**Fig. 3 | Copper ion redox mechanism in Cu − Cl electrolyte. a** CV curves obtained at a scan rate of 1 mV s⁻¹. **b** Corresponding ex situ XRD patterns. **c** Corresponding ex situ UV-Vis spectra. **d** Corresponding ex situ XPS spectra. **e** Corresponding ex situ XAES spectra. **f** Energy profiles of possible reaction pathways. The electrochemical test was conducted in a three-electrode cell consisting of a carbon felt working electrode, a Pt plate (1 × 1 cm) counter electrode, and an SCE reference electrode.

performed at scan rates from 1 to 10 mV s⁻¹ (Fig. 4a, b). As the scan rate was increased, the stability of the copper ions' redox behaviours increased for both coordination environments. The charge storage kinetics were analysed using the equation $I = av^b$, where $I$ and $v$ are the peak current and scan rate, respectively, and $a$ is a constant[47]. Generally, a $b$ value close to 0.5 suggests that electrochemical behaviour is a diffusion-controlled process, whereas a $b$ value approaching 1 implies a non-diffusion capacitive-controlled process. The linear fitted $b$ values calculated from the two redox pairs under chlorine coordination were 0.46 ± 0.01 and 0.77 ± 0.01 (Fig. 4c), indicating that the electrochemical reactions were controlled by a mixed diffusion–capacitive co-controlled process. However, the $b$ value for water coordination was 0.30 ± 0.01, indicating a slow diffusion-controlled process. The diffusion contribution ($k_2v^{1/2}$) and non-diffusion capacitive contribution ($k_1v$) at various scan rates could be calculated using the equation $i(V) = k_1v + k_2v^{1/2}$, where $k_1$ and $k_2$ are constants, $v$ is the scan rate, and $i(V)$ is the current density[48]. As the scan rate was increased from 1 to 10 mV s⁻¹, the capacitive contribution increased from 49.3% to 75.4% for the chlorine coordination and increased from 28.1% to 54.2% for the water coordination (Fig. 4d). The relevant CV curves are shown in Supplementary Figs. 13 and 14. The higher capacitive contribution for the chlorine coordination signifies that the reaction kinetics were faster under this condition, which can be attributed to the strengthened interaction between copper and chlorine ions that facilitated efficient charge transfer processes. This finding suggests that the copper negative electrode tended to have higher rate capability than it did under water coordination.

Galvanostatic charge–discharge (GCD) curves obtained under chlorine and water coordination for various current densities are directly compared in Fig. 4e, f. The copper negative electrode in the Cu −Cl electrolyte exhibited an areal capacity of 0.31 mAh cm⁻² at the current density of 10 mA cm⁻², whereas it exhibited a lower areal capacity of 0.2 mAh cm⁻² in the Cu−H₂O electrolyte, indicating higher utilisation of copper ions in the chlorine coordination environment. As illustrated in Fig. 4g, the copper negative electrode under chlorine coordination had higher rate capability than that under water coordination. Specifically, at the current density of 40 mA cm⁻², the copper negative electrode had large capacity of 0.23 mAh cm⁻² under chlorine coordination but a capacity of only 0.12 mAh cm⁻² under water coordination. Under chlorine coordination, the copper negative electrode exhibited excellent stability, with 99% capacity retention after 5000 cycles at a current density of 20 mA cm⁻², whereas under water coordination, the copper negative electrode failed after 4500 cycles (Fig. 4h, and the corresponding Coulombic efficiency curves are rescaled in Supplementary Fig. 15). The greater cycle stability under chlorine coordination was attributable to the more uniform deposition of metallic copper on the carbon felt, which meant that the deposition−dissolution was more reversible.

These results revealed that chlorine coordination can result in the following virtues of a copper negative electrode: (i) an approximately 0.3 V lower negative electrode potential, (ii) higher utilisation of copper ions, (iii) faster electrochemical kinetics, and (iv) better cycle stability.

## The electrochemical behaviours of positive electrode
A suitable positive electrode was required to further assess the effectiveness of chlorine coordination in a full-cell configuration. Considering the presence of excess chloride ions (Cl⁻) in the Cu−Cl electrolyte, we decided to directly employ the Cl⁻/Cl⁰ redox reaction at the positive electrode to construct a Cu-Cl₂ battery system, and the positive electrode Cl⁻/Cl⁰ reaction was thus investigated in detail. CV was performed from 0.5 to 1.2 V (vs. SCE) in both electrolytes when

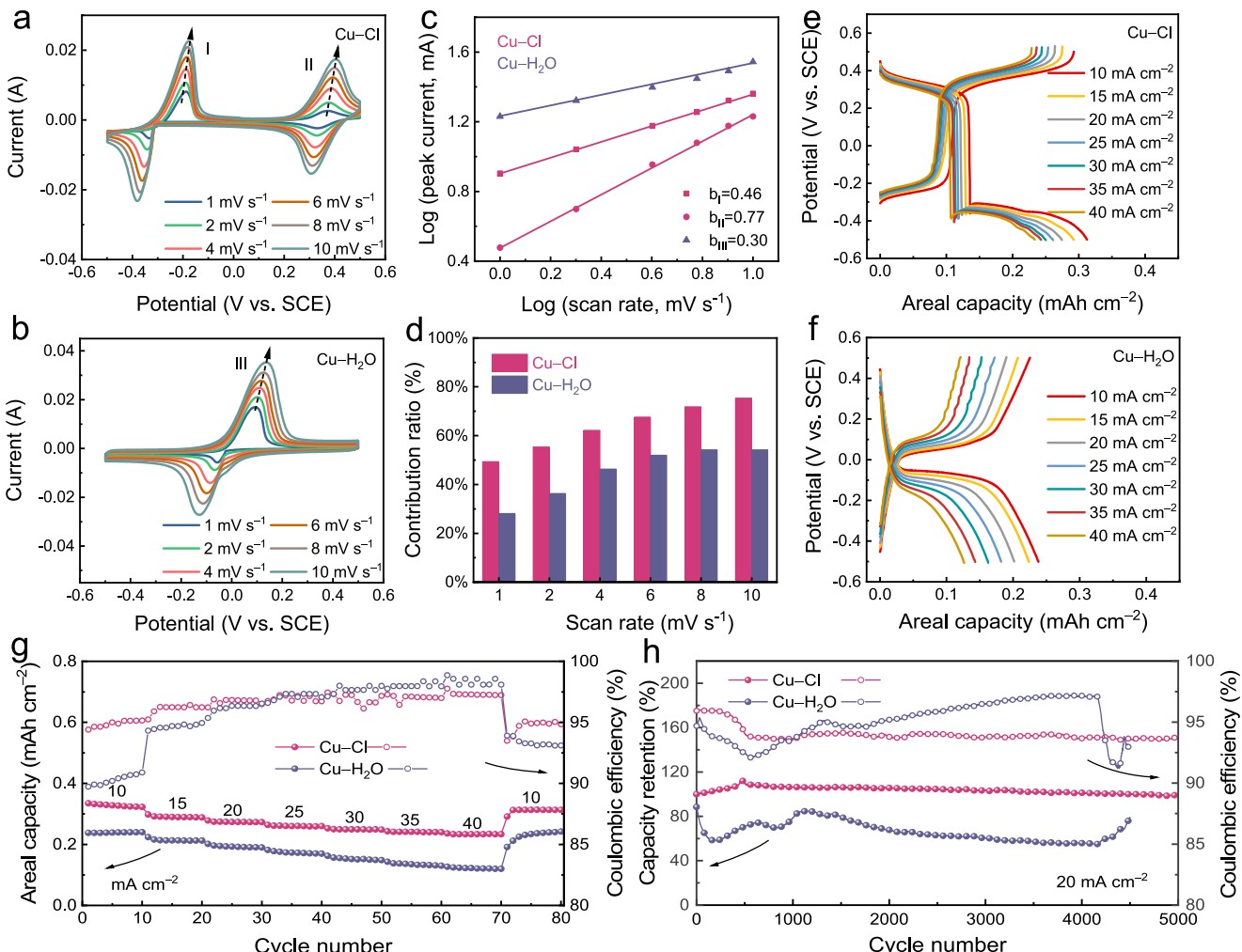

**Fig. 4 | The electrochemical performance of Cu negative electrode.** CV curves of the Cu negative electrode at different scan rates in (**a**) Cu−Cl and (**b**) Cu−H$_2$O electrolytes. **c** The b values corresponding to the three oxidation peaks in **a** and **b**. **d** Capacitive contributions at various scan rates. **e** GCD curves in Cu−Cl electrolyte at various current densities. **f** GCD curves in Cu−H$_2$O electrolyte at various current densities. **g** Rate performance in Cu−Cl and Cu−H$_2$O electrolytes. **h** Long-term stability in Cu−Cl and Cu−H$_2$O electrolytes at 20 mA cm$^{-2}$. The electrochemical tests were conducted in a three-electrode cell consisting of a carbon felt working electrode, a Pt plate (1 × 1 cm) counter electrode, and an SCE reference electrode.

using a standard three-electrode system. A Ketjen black (KJB) electrode was used as the working electrode due to its high chlorine adsorption activity. A reduction peak clearly appeared at 1.1 V (vs. SCE) when the Cu-Cl electrolyte was employed, whereas only a capacitive-type curve appeared for the Cu-H$_2$O electrolyte, indicating the feasibility of using Cl$^-$/Cl$_2$ as the positive electrode reaction (Fig. 5a). In agreement with the CV curve, the charge−discharge profile of the Cl$^-$/Cl$_2$ reaction contained a discharge plateau at 1.02 V (vs. SCE) when the specific current was 10 A g$^{-1}$ (Fig. 5b). The KJB electrode was discovered to exhibit poor cycle stability, with a low capacity retention of 21% after 550 cycles. Therefore, 0.5 M HCl was incorporated into the Cu−Cl electrolyte (pH ≈ 1.3) to improve the stability of the KJB electrode, and the optimal cycle stability with 88% capacity retention after 5000 cycles was achieved (Fig. 5c). UV-Vis spectra were obtained to gain insight into the mechanism underlying this cycle stability optimisation. To exclude copper ion interference, UV-Vis spectra were measured at various charge potentials in the electrolyte solution comprising 4 M NaCl and 0.5 M HCl (Fig. 5d) and that comprising 4 M NaCl (Fig. 5e). When the voltage was higher than 1.1 V, a peak was present at 324 nm for the electrolyte containing 0.5 M HCl, indicating the presence of molecular chlorine (Cl$_2$)[49,50]. The intensity of the peak increased with the potential, which further confirmed that the redox reaction

belonged to Cl$^-$/Cl$_2$. When the pure NaCl electrolyte was used, one peak was found (at 290 nm) that corresponded to the ClO$^-$; this was attributable to the hydrolysis of molecular chlorine (Cl$_2$ + H$_2$O ↔ HCl + HClO)[51,52]. Therefore, molecular chlorine was concluded to tend to react with water, causing the loss of active substances in the electrolyte, whereas when HCl was present, the hydrolysis of molecular chlorine was suppressed, preventing the formation of HClO and enhancing the cycling stability. Furthermore, the addition of HCl effectively suppressed the competitive oxygen evolution reaction (OER). Real-time monitoring of oxygen production using an in situ gas chromatography system demonstrated significant reduction in the oxygen production rate (from 0.1 ml h$^{-1}$ down to approximately 0.03 ml h$^{-1}$) upon the addition of 0.5 M HCl, as shown in Supplementary Fig. 16.

The electrochemical behaviours of the positive electrode Cl$^-$/Cl$_2$ reaction were evaluated in detail. The CV curve obtained at the positive electrode had the same approximate shape for scan rates from 1 to 10 mV s$^{-1}$ (Fig. 5f), indicating stable redox behaviour. The charge−discharge curves obtained at various specific currents are presented in Fig. 5g and reveal specific capacity of 160 mAh g$^{-1}$ at 5 A g$^{-1}$. Remarkably, when the specific current was increased to 30 A g$^{-1}$, almost no decrease in specific capacity was observed

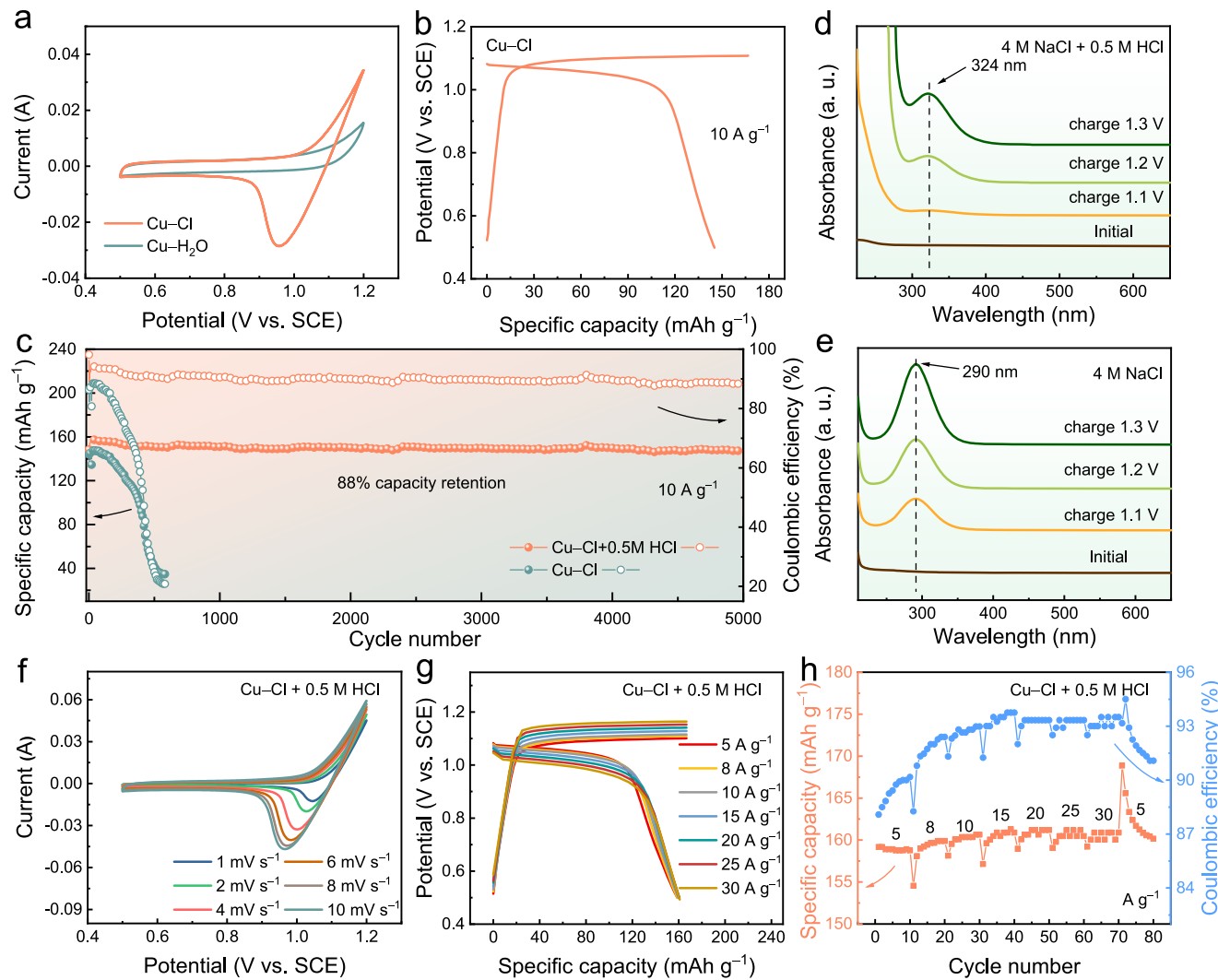

**Fig. 5 | The electrochemical behaviours of positive electrode. a** CV curves obtained at a scan rate of 10 mV s⁻¹ when using Cu-Cl and Cu-H₂O electrolytes. **b** GCD curve of the Cl₂ positive electrode at 10 A g⁻¹. **c** Long-term stability of the Cl₂ positive electrode in Cu-Cl and Cu-H₂O electrolytes at 10 A g⁻¹. UV-Vis spectra obtained at various potentials in a solution comprising (**d**) 4 M NaCl and 0.5 M HCl and (**e**) 4 M NaCl. **f** CV curves. **g** GCD curves. **h** Rate performance of the Cl₂ positive electrode in a solution comprising 4 M NaCl and 0.5 M HCl. The electrochemical tests were conducted in a three-electrode cell consisting of a KJB working electrode, a Pt plate (1 × 1 cm) counter electrode, and an SCE reference electrode.

compared with the decrease that occurred at lower specific currents (Fig. 5h), revealing excellent positive electrode rate capability. In addition, the effect of HCl on the copper negative electrode was excluded by analysing the redox behaviour in the Cu-Cl electrolyte and the electrolyte containing Cu-Cl and 0.5 M HCl. As revealed by the CV and GCD curves (Supplementary Figs. 17 and 18, respectively), the presence of HCl did not change the redox potential of the copper negative electrode, and the specific capacity was only slightly increased.

## The electrochemical properties of the Cu-Cl₂ full cell

With the negative and positive electrode electrochemical properties well clarified, the Cu-Cl₂ full-cell performance was examined. The full-cell configuration is illustrated in Fig. 6a. As revealed by CV curves, the positive electrode redox potential was observed at approximately 1.1 V (vs. SCE), corresponding to the Cl⁻/Cl₂ reaction. Regarding the negative electrode potential, two redox pairs were discovered at approximately −0.28 and 0.36 V (vs. SCE), respectively, which were associated with the Cu⁺/Cu⁰ and Cu²⁺/Cu⁺ reactions. On the basis of this difference in potential, the full cell was expected to yield two charge–discharge platforms at approximately

1.4 and 0.7 V. The charge–discharge profiles of the full cell were validated using the CV and GCD results presented in Fig. 6b, c. The Cu-Cl₂ battery demonstrated favourable rate capability (Fig. 6d) and achieved a large discharge capacity of 127 mAh g⁻¹ (based on the mass of KJB) even at a high specific current of 20 A g⁻¹. Moreover, the full cell exhibited a capacity retention rate of 83.5% after 5000 cycles for 20 A g⁻¹ (Fig. 6e), indicating excellent cycle stability. Even after 10,000 cycles, the retention rate was still high at 77.4%. As expected, benefiting from chlorine coordination, the Cu-Cl₂ full cell delivered a specific capacity of 162 mAh g⁻¹ (based on the mass of KJB) at 5 A g⁻¹. It also presented a high discharge plateau at 1.3 V the specific values are compared with literature reports in Supplementary Table 1). Despite the promising performance of the designed Cu-Cl₂ battery system, it is essential to acknowledge its current limitations. One of the primary concerns associated with this system pertains to the involvement of the volatile and hazardous Cl₂ gas, and implementing safety measures such as ventilation and containment systems may effectively mitigate the risk of Cl₂ release. Additionally, the volatility of Cl₂ also raises concerns regarding the self-discharge of the system, potentially reducing the efficiency and performance of the battery. These considerations are of paramount importance

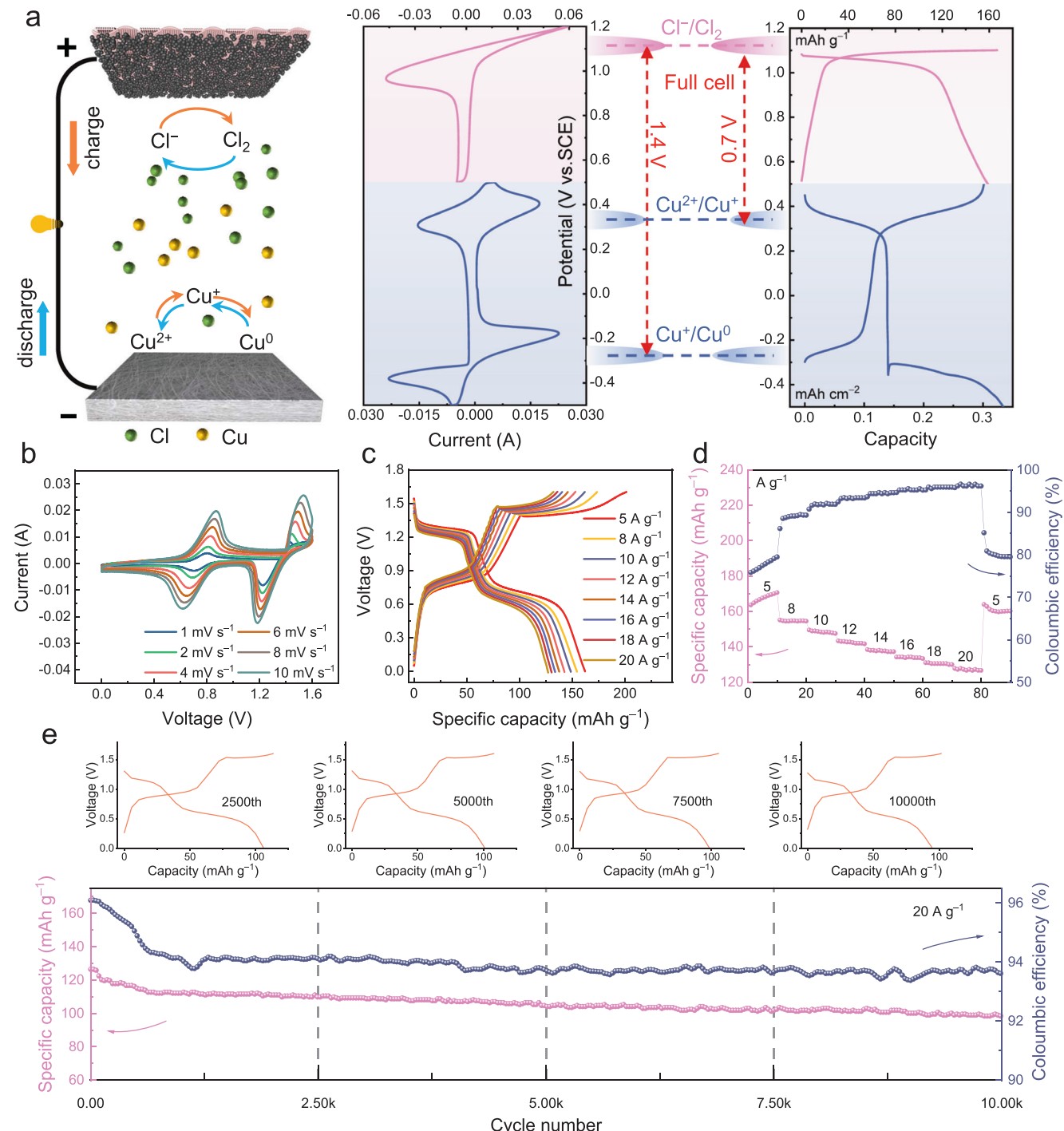

**Fig. 6 | The electrochemical properties of the Cu-Cl₂ full cell. a** Schematic of the Cu-Cl₂ full cell. **b** CV curves obtained at various scan rates. **c** GCD curves obtained at different specific currents of the Cu-Cl₂ full cell. **d** Rate performance. **e** Long-term stability of the Cu-Cl₂ full cell at 20 A g⁻¹. The electrochemical tests were conducted in a two-electrode cell consisting of a carbon felt negative electrode and a KJB positive electrode.

for the practical implementation of the Cu-Cl₂ batteries and warrant further investigation in future research.

## Discussion

We thoroughly investigated the effect of chlorine coordination on the redox potential of copper to obtain high-voltage aqueous copper-based batteries. Due to its complexation with Cu⁺, Cl⁻ can stabilise Cu⁺ in aqueous electrolyte to enable a redox reaction of Cu⁺/Cu⁰ to occur at a potential -0.3 V lower than that of the conventional Cu²⁺/Cu⁰ reaction. Compared with that under water coordination, the copper

negative electrode under chlorine coordination also exhibited higher areal capacity of 0.31 mAh cm⁻² at 10 mA cm⁻², faster redox kinetics, and higher cycle stability, achieving 99% capacity retention after 5000 cycles at 20 mA cm⁻². Finally, a Cu-Cl₂ battery was constructed in situ by employing Cl⁻/Cl₂ as the positive electrode reaction. The copper battery provided a distinctively high discharge plateau at 1.3 V and had discharge capacity of 162 mAh g⁻¹ at 5 A g⁻¹ (based on the mass of KJB). It also demonstrated excellent rate capability and favourable cycle stability, retaining 77.4% of its initial capacity over 10,000 cycles. The strategy used in this study—regulating the coordination environment—

resulted in lowered negative electrode potential, improved electrochemical kinetics, and enhanced cycle stability. These findings can shed light on future development of high-performance aqueous batteries.

## Methods

### Materials

All chemical regents and materials used in this work are commercially available and used without further purification. Sodium chloride (NaCl), potassium chloride (KCl), calcium chloride (CaCl$_2$·2H$_2$O) and sodium sulfate (Na$_2$SO$_4$) were purchased from Aladdin. Copper chloride (CuCl$_2$·2H$_2$O) and copper sulfate (CuSO$_4$·5H$_2$O) were purchased from Macklin. Ketjen Black (ECP600JD) were purchased from Suzhou Slnero Technologu Co., Ltd. All electrolytes were prepared using deionized water (resistance of 18.2 MΩ).

### Electrochemical measurements

To prepare the Ketjen Black (KJB) electrode, ninety percent by weight of Ketjen Black, five percent by weight of sodium carboxymethyl cellulose (CMC) and five percent by weight of styrene-butadiene rubber (SBR) were mixed and grinded with deionized water to make a homogeneous slurry. Then the mixed paste was cast on carbon cloth current collectors. The active mass loading was about 2 mg cm$^{-2}$. Carbon cloth was selected as the positive electrode collector due to its superior conductivity, chemical stability, and high specific surface area. Notably, its impact on battery capacity was found to be negligible, as evidenced in Supplementary Fig. 19. The cyclic voltammetry (CV) tests were operated with a CHI 700E electrochemical workstation, while the galvanostatic charge–discharge rate capability and battery cycling performance were carried out on a Neware battery test system. The electrochemical performance of the half-cell was tested in a three-electrode cell, where Pt plate (1 × 1 cm) was employed as the counter electrode and saturated calomel electrode (SCE) was used as the reference electrode. The electrochemical performance of the full cell was tested using KJB electrode as the positive electrode and carbon felt as negative electrode. All electrochemical tests were conducted in a controlled environmental chamber at 25 ± 0.5 °C. In all, 5 mL of each electrolyte was added to the three-electrode cell and full-cell tests. The working electrode dimensions in a three-electrode cell, and the positive/negative electrode dimensions in a full cell, are both 1 × 1 cm. At least 5 cells were tested for each electrochemical test (more details please see Supplementary Note 1). The negative electrode, comprising carbon felt without active substance, was evaluated for specific capacity and current density based on its projected area. The specific capacity and specific current of the positive electrode and the full cell were determined using the mass of Ketjen black.

The stability constants were computed using the equation lgβ$_n$ = [CuX$_n$]/([Cu$^+$] [X$^-$])[53], where [CuX$_n$] represents the concentration of the complex, [Cu$^+$] represents the concentration of Cu$^+$ ions, [X$^-$] represents the concentration of halide ions, and n represents the stoichiometry of the complex.

### Physicochemical characterisations

The XRD patterns were recorded on a Mini Flex 600 X-ray diffractometer (Rigaku) from 10 ° to 80 ° using a Cu Kα radiation (λ = 1.5418 Å). The scanning electron microscope (SEM) images were taken on a VEGA3 SEM. The XPS and XAES were measured on the Thermo Fisher ESCALAB 250Xi system with a monochromatic Al Kα source (1486.6 eV), using the C 1 s = 284.8 eV to calibrate the binding energy value of each element. The UV−VIS spectra were collected on a Lambda 365 UV−VIS spectrophotometer. 4 M NaCl was used as the blank sample to deduct its absorption, and the concentration of CuCl$_2$ were diluted to around 10 mM. The SERS spectra were recorded using the Raman spectrometer (Thermo Fischer DXR). NMR spectra were collected with Bruker AVANCE III HD 400 MHz.

The oxygen production monitored on an Agilent 7820 A gas chromatography.

### Computational method

Molecular dynamics simulations were conducted using the Forcite Package with the COMPASS III force field[30]. Atom-based methods were employed to analyse electrostatic and van der Waals interactions. Geometry optimisation using the smart method was performed to obtain a reasonable interaction configuration, with convergence criteria of 1.0 × 10$^{-3}$ kcal mol$^{-1}$ for energy and 1.0 × 10$^{-2}$ kcal mol$^{-1}$ Å$^{-1}$ for forces. The simulations were equilibrated under constant pressure and temperature (NPT ensemble) for 1 ns at a room temperature of 298.0 K and atmospheric pressure of 1.013 × 10$^{-4}$ GPa. Equilibrated simulations were then run at a constant NVT ensemble for 5 ns to collect and analyse data. Quantum chemistry calculations were carried out using the Gaussian 09 software package[54]. Geometries, energies, and frequencies of all stationary points (reactants, products) were obtained using density functional theory with the B3LYP method and the SDD basis set for Cu and 6-311 + G basis set for H, O, and Cl[55,56].

### Reporting summary

Further information on research design is available in the Nature Portfolio Reporting Summary linked to this article.

## Data availability

The data that support the findings of this study are available from the corresponding author upon request.

## Code availability

The codes that support the findings of this study are available from the corresponding author upon request.

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

## Acknowledgements

The authors acknowledge the financial support provided by the National Natural Science Foundation of China (52103089 awarded to Z.L. and 22005207 awarded to H.L.), Guangdong Basic and Applied Basic Research Foundation (2023A1515012120 awarded to Z.L., 2022A0505050015 and 2021B1515120004 awarded to H.L.) and the Shenzhen Science and Technology Program (JCYJ20220531100815035 awarded to Z.L.). The authors also thank the Instrumental Analysis

Center of Shenzhen University (Lihu Campus) for their assistance with SEM characterisation.

## Author contributions

Z.L., H.L. and C.Z. conceived and supervised the project. H.L. and X.Z. designed the experiments. X.Z. performed the experiments. X.Z., H.W., S.L., B.R., J.J., G.Q., H.L. and G.L. analysed and interpreted the results. X.Z. wrote the paper. Z.L., H.L., C.Z. and G.C. revised the paper. All authors discussed the results and commented on the manuscript.

## Competing interests

The authors declare no competing interests.
