## [Peer Review File · Nature Communications]

Reviewer #1 (Remarks to the Author):

In the manuscript entitled 'Manipulating coordination environment enables a high-voltage aqueous copper-based battery,' Zhang et al. described the effect of the halogen anions on the redox behavior of copper ions in aqueous electrolytes. Their main findings related to the influence of the halogen anion type on the redox of metallic Cu, and the presence of an intermediate form of Cu⁺ is interesting, particularly from a scientific point of view. At moment, I don't think that this system can be practical (see my comment below), however, this system demonstrates an important concept related to designing principles of aqueous battery systems. Overall, I support the publication of this manuscript in Nature Communications, however, the following issue should be first addressed:

Major comments

Line 106 How the stability constance was calculated? Please add more details

Line 138 The charging mechanism proposed by the authors includes a transformation of CuCl to an intermediate phase of CuCl₃²⁻. An alternative explanation is a conversion of CuCl₂ to NaCl and CuCl (see for example 10.1149/2.0461904jes). While such a situation occurs when the starting electrode materials consist of CuCl₂ particles, the participation of solid CuCl₂ during the cycling, particularly in saturated solutions of CuCl₂ and NaCl is very reasonable (in my view the potential of the redox peaks in the solid reaction are similar the following work). Although based on the XRD there is no evidence for the formation of CuCl, I am afraid that the intensity of this phase is insufficient to detect in XRD. Therefore the authors should provide more solid evidence for the proposed charging mechanism. In this context, Raman spectroscopy XPS of the solidified electrolyte as well as theoretical simulation may give a hint for the occurring reactions.

Line 171 Two different XPS spectra should be presented –after charging to 0.3V (formation of the intermediate phase), and after Cu deposition. This will give appropriate information about the changes in the electrode surface as a function of the applied potential.

Line 195 To which potential the b values are correlated? Please show the full CV response from which the contribution ratio (Figure 3d was calculated)

A general note While the performance of the full cell is presently nice, the author should also indicate what are the practical limitations of this system. Clearly, as Cl₂ is a volatile gas, such a system can not be operated at very slow charging rates which obviously results in the generation of a significant amount of hazardous Cl₂ gases. In addition, I expect that the self-discharge of the

system will be dramatically fast (as Cl₂ will not remain on the surface for a long time). Nevertheless, I think that from a scientific point of view, this manuscript is very interesting. Yet as scientists, we have to point out what are the limitations and the challenges of this system. This in my view will not reduce the importance and effectiveness of the work, but oppositely will be highly appreciated by readers and will help to further develop aqueous batteries for energy storage applications.

Minor comments

Line 46 – Please add also Mn ion battery which also starts to attract attention. The following references are recommended (10.1021/acseenergylett.2c02242, 10.1002/ange.202200809)

Line 96 As far as I know, NaF can not reach 4M in an aqueous medium. What is the actual concentration of the saturated NaF solution?

Line 159 It seems that the reference number 'jumped' into the oxidation number notation of the Cu.

Reviewer #2 (Remarks to the Author):

The work suggested through manipulating coordination chemistry of Cu-Cl₂ for a high voltage aqueous battery is conceptually good potential of interest. Despite the authors have shown valuable results on electrochemical performance/properties, there is a huge gap in explaining the coordination chemistry of Cu-Cl, which is important as mentioned in the title of the manuscript, resulting lack of evidence for proving their origin of formations. The reviewer cannot recommend this manuscript for publication in Nature Communications. Few following comments can help the authors.

(i) There is an empty space to explain the Cu-Cl complex coordination, questioning basic chemistry behind solvation of salts in water. What is the coordination chemistry of CuCl₂. 2H₂O + 4M NaCl in aqueous medium? Wondering that it is not a very high concentrated solution like water-in-salt solution to completely repel all water molecules surrounded by Cu for coordination, then how the coordination has derived in absence of H₂O molecules in a large pool of aqueous solution?

(ii) What is the structure and coordination numbers of each intermediate products formed during reaction? For example, what would be the structure/coordination number of [CuCl₃]²⁻ as an intermediate species formed during reduction? There is no structural evidence. Explain in detail. Why not the coordination would be like [CuCl₂(H₂O)]⁻, in which Cu stands at +1 oxidation state.

(iii) To establish the electrochemical mechanisms further, analyzing the residual electrolytes and the carbon electrodes (both carbon felt/cloth and KJB) available during the electrochemical reactions for both cases, are strongly required. There is a need of strong evidence to confirm the $\text{Cu}^{2+}/\text{Cu}^{+}$ and $\text{Cu}^{+}/\text{Cu}^0$ through valuable spectroscopy techniques to support the UV-Vis analysis. Find a suitable analysis that helpful to understand both oxidation states and coordination chemistries of Cu in both electrolytes (CuCl_2 and CuSO_4). Hint. Resonance spectroscopy.

(iv) How UV-Vis analysis have been measured? With or without dilution? In such a high concentration of electrolyte, dilution is commonly employed to avoid peak amplitude saturation and in doing so, the coordination chemistry will be changed. Explain briefly.

(v) How to control the further dissociation of HClO into hypochlorite (OCl^-), resulting in formation of H^+ in presence of strong 0.5 M HCl added electrolyte? What would be a pH value? The above said dissociation reaction could be spontaneous reaction ($\text{Cl}_2 + 2\text{H}_2\text{O} \rightarrow \text{HOCl} \rightarrow \text{H}^+ + \text{OCl}^-$)!

(vi) Why the cathode reaction of Cl^-/ClO has been shown in terms of gram current density (mA g^{-1}) rather than areal density (mA cm^{-2}) as given for anode reaction? Similarly, the fully developed cells too represent by mA g^{-1} .

Additional comments

(vii) Typos errors: line 159 $\text{Cu}^{2+}25$. May the reference number interfere?

(viii) Fig. 3h, y-axis capacity retention (%) might be wrong.

Reviewer #3 (Remarks to the Author):

This manuscript reported a coordination strategy for reducing the intrinsic anode redox potential in aqueous copper-based batteries to improve the operating voltage. The stabilized Cu^+ ions by chlorine coordination present a redox potential approximately 0.3 V lower than that for $\text{Cu}^{2+}/\text{Cu}^0$. The Cl^-/ClO redox reaction was further employed as the cathode to achieve a novel aqueous $\text{Cu}-\text{Cl}_2$ battery with a high discharge voltage of 1.3 V and good stability. Overall, it is an interesting work and may provide a new insight into how the coordination environment influences the anode potential and the electrochemical performance for aqueous batteries. Thus, I recommend its publication on Nature Communications after addressing the following minor issues.

1. Why did the authors select the 4 M NaCl and 0.05 M CuCl₂ as the electrolyte, which was not mentioned in the manuscript? How about its performance with other ratios? The authors should explain their choice on electrolyte more profoundly.
2. The authors used carbon cloth as the anode and actually constructed an anode-free Cu-Cl₂ battery. How does it perform when metallic copper is used as the anode? What are the advantages of carbon cloth in comparison?
3. Fig. 3g, the copper anode under chlorine coordination shows higher rate capability than that under water coordination, which means the faster electrochemical kinetics for chlorine coordination. What is the specific reason? Please clarify this point.
4. The authors used carbon cloth as a cathodic current collector. Can it adsorb molecular chlorine to contribute to the capacity? What are the criteria for selecting the current collector?
5. Besides the hydrolysis of molecular chlorine ($\text{Cl}_2 + \text{H}_2\text{O} \leftrightarrow \text{HCl} + \text{HClO}$), the competing oxygen evolution reaction (OER) should also receive attention. The authors need to evaluate the oxygen evolution reaction (OER) process after the incorporation of HCl. In this case, the suppression of side reactions can be well studied.
6. The Cu-Cl₂ full cell delivered high energy and power densities of 141.2 Wh kg⁻¹ and 4236 W kg⁻¹, respectively. How does it calculate? Is it based on the mass of both cathode and anode active materials?
7. There are some minor issues that should be addressed, for example, Page 4, line 86, add "a" in front of high discharge voltage; Page 6, line 137, replace "was" by "were".
Page 9, line 198, replace " i(v)" by " i(V)".

Response to Reviewers

Dear Reviewers:

Thanks a lot for your constructive comments and suggestions concerning our manuscript entitled "*Manipulating coordination environment enables a high-voltage aqueous copper-based battery*". Your feedbacks are all valuable to help improve our manuscript. We have studied your comments very carefully and accordingly revised our manuscript, hoping that it can better meet the high standards of *Nature Communications*. The comments have been addressed point-by-point below, and the related revisions are highlighted in yellow in the manuscript as well as in the Supplementary Information.

To Reviewer #1

Overall comment:

In the manuscript entitled ‘Manipulating coordination environment enables a high-voltage aqueous copper-based battery,’ Zhang et al. described the effect of the halogen anions on the redox behavior of copper ions in aqueous electrolytes. Their main findings related to the influence of the halogen anion type on the redox of metallic Cu, and the presence of an intermediate form of Cu^+ is interesting, particularly from a scientific point of view. At moment, I don’t think that this system can be practical (see my comment below), however, this system demonstrates an important concept related to designing principles of aqueous battery systems. Overall, I support the publication of this manuscript in Nature Communications, however, the following issue should be first addressed:

Response:

Thank you for your constructive feedback and endorsement of our manuscript. We acknowledge your recognition of the scientific significance of our findings concerning the role of halogen anions on copper ion redox behavior in aqueous electrolytes, especially the discovery of an intermediate form of Cu^+ .

We acknowledge the current limitations of our system, yet emphasize that our research contributes valuable insights towards the formulation of new design principles for aqueous battery systems. This could potentially facilitate the development of more practical and efficient energy storage solutions. In response to your suggestions, we have incorporated more mechanism analyses into our manuscript.

Major comments

1. Line 106 How the stability constant was calculated? Please add more details

Response:

Thank you for your valuable query. The stability constants presented in this study

were referenced from the Lange Handbook of Chemistry, and appropriate citation has been added in the text (Table 1.75, Table 1.76, *Lange's Handbook of Chemistry*, 16th ed., McGraw-Hill Professional Publishing, New York, USA, 2005). The stability constant of a coordination compound is a crucial parameter that denotes the strength of the bond between a metal ion and its ligands. It is defined as the equilibrium constant for the formation of a complex in a solution, involving a metal ion and its ligands. As the stability constant increases, the complex becomes more stable.

Specifically, for Cu⁺ complexes with halide ions, the stability constants were computed using a well-established equation shown below (*J. Solution Chem.* 1990, 19, 375-390):

$$\lg\beta_n = [\text{CuX}_n]/([\text{Cu}^+][\text{X}^-]^n)$$

where [CuX_n] signifies the concentration of the complex, [Cu⁺] denotes the concentration of Cu⁺ ions, [X⁻] represents the concentration of halide ions, and *n* is the stoichiometry of the complex.

We have incorporated the detailed calculation into the "Methods" section. Please refer to the highlighted revision made on page 19.

2. Line 138 The charging mechanism proposed by the authors includes a transformation of CuCl to an intermediate phase of CuCl₃²⁻. An alternative explanation is a conversation of CuCl₂ to NaCl and CuCl (see for example 10.1149/2.0461904jes). While such a situation occurs when the starting electrode materials consist of CuCl₂ particles, the participation of solid CuCl₂ during the cycling, particularly in saturated solutions of CuCl₂ and NaCl is very reasonable (in my view the potential of the redox peaks in the solid reaction are similar the following work). Although based on the XRD there is no evidence for the formation of CuCl, I am afraid that the intensity of this phase is insufficient to detect in XRD. Therefore the authors should provide more solid evidence for the proposed charging mechanism. In this context, Raman spectroscopy XPS of the solidified electrolyte as well as theoretical simulation may give a hint for the occurring reactions.

Response:

We appreciate your insightful comments and alternative explanation for our proposed charging mechanism. We understand your concern about the potential participation of solid CuCl_2 during the cycling process. We concur that the absence of CuCl detection in the XRD analysis may be attributed not only to its low content but also to its amorphous nature. In response to this, we performed additional *ex-situ* Raman and *ex-situ* XPS analysis on the reacted carbon cloth, which was coated with activated carbon that possesses strong adsorption capacity, to further support our findings.

In the Raman spectra shown in Fig. 1, the peaks observed at 215 and 232 cm^{-1} are attributed to the characteristic Cu-Cl bonds, while the peak at 247 cm^{-1} corresponds to the bending vibration of Cu-Cl bonds (*J. Electrochem. Soc.*, 2018, 165, C375; *Vib. Spectrosc.*, 2018, 99, 1-6; *Neues Jahrb. für Mineral. Abhandlungen*, 2003, 178, 197-215). As highlighted in Fig. R1b, the Cu-Cl bending vibration peak emerged during the discharge process, but gradually weakened upon charging.

Fig. R1. Copper ion electrochemical behaviors under chlorine coordination. (a) Cyclic voltammetry (CV) curve obtained at a scan rate of 1 mV s^{-1} for Cu-Cl electrolyte. (b) *Ex-situ* Raman spectra corresponding to different potentials marked in (a).

Yet, we acknowledge the difficulty in distinguishing between CuCl and CuCl_2 based solely on Raman spectroscopy due to the similar bonding environments, which lead to overlapping vibrational modes and frequencies. To resolve this, we employed X-ray photoelectron spectroscopy (XPS) and X-ray excited Auger Electron

Spectroscopy (XAES) to further distinguish between Cu^0 and Cu^+ species.

The XPS spectra revealed distinct peaks corresponding to different copper species. As shown in Fig. R2a, the $\text{Cu } 2p_{3/2}$ and $\text{Cu } 2p_{1/2}$ peaks were observed at approximately 935.2 and 955 eV, respectively, indicating the presence of Cu^{2+} species. While the $\text{Cu } 2p_{3/2}$ and $\text{Cu } 2p_{1/2}$ peaks at 932.3 and 952 eV, respectively, were attributed to Cu^+ and/or metallic Cu^0 species (*Adv. Mater.*, 2022, 34, 2205229; *Angew. Chem. Int. ed.*, 2022, 134, e202212191). Thus, these peaks detected at II, III, and IV states indicate the formation of Cu^+ and/or Cu^0 species during discharging. Nevertheless, due to the small difference in binding energies, it is still challenging to distinguish between the chemical states of zero-valent copper (Cu^0) and univalent copper (Cu^+) based solely on these XPS spectra (*J. Am. Chem. Soc.*, 2013, 135, 14032-14035; *J. Phys. Chem. C*, 2008, 112, 1101-1108). To gain further insights, XAES characterizations were conducted. As shown in Fig. R2b, the Cu LMM peak analysis reveals distinct peaks corresponding to different copper species. Specifically, the peak centered around 918.5 eV was consistent with metallic Cu^0 , while the peak at 915.5 eV was attributed to Cu^+ . Additionally, the peak at approximately 917 eV was associated with Cu^{2+} (*ACS Sustain. Chem. Eng.*, 2022, 10, 15958-15967; *Appl. Surf. Sci.*, 2021, 544, 148915).

Fig. R2. Analysis of copper valence state. (a) *Ex-situ* XPS spectra corresponding to different potentials marked in Fig. R1a. (b) *Ex-situ* XAES spectra corresponding to

different potentials marked in Fig. R1a.

Combining the previous X-ray diffraction (XRD) and ultraviolet-visible spectroscopy (UV-Vis) analysis, these additional *ex-situ* characterizations provide robust evidence for the reversible redox reaction of the Cu anode: in the initial state (I), the valence state of copper (Cu) is primarily +2. As the CV negative scan proceeds (I → III), the predominant Cu²⁺ species undergoes partial reduction to Cu⁺/Cu⁰. Subsequently, during the subsequent positive scan process (III → V), the Cu⁰ species is oxidized back to Cu⁺ and further to Cu²⁺.

Consequently, we propose the following electrochemical process to describe the proposed mechanism:

Furthermore, to substantiate our proposed mechanism, we carried out theoretical simulations to investigate the solvation structure and reaction pathways of Cu–H₂O and Cu–Cl electrolytes. These simulations were beneficial in shedding light on the distinct characteristics of each electrolyte system. As shown in Fig. R3a, in the Cu–H₂O electrolyte, the coordination of copper ions with six water molecules resulted in the formation of a [Cu(H₂O)₆]²⁺ complex. The simulations demonstrate a direct conversion of [Cu(H₂O)₆]²⁺ to Cu(H₂O)₆, as the proposed intermediate species [Cu(H₂O)₆]⁺ exhibited higher energy (5.1 eV) and would readily transform into the more stable Cu(H₂O)₆ (2.6 eV). Conversely, the solvation structure of copper ions in the Cu–Cl electrolyte displayed greater complexity, where the copper-water-chloride complex entailed approximately 1.8 chloride ions and 4.2 water molecules (Fig. R3b). The simulation outcomes unveil a reaction pathway commencing with the formation of CuCl(H₂O)₄ from CuCl₂(H₂O)₄, necessitating the surmounting of a 0.9 eV thermodynamic energy barrier (Fig. R3c). The formed CuCl(H₂O)₄ subsequently underwent further reactions, culminating in the ultimate production of Cu(H₂O)₄. Notably, these simulation findings closely align with our CV data, offering robust support for the proposed reaction mechanism.

Fig. R3 Theoretical simulation of the coordination chemistry and reaction pathways in the Cu-H₂O and Cu-Cl electrolytes. (a) Radical distribution function (RDF) of the Cu-H₂O electrolyte, (b) RDF of the Cu-Cl electrolyte. (The solid lines are the radial distribution functions, and the dotted lines are the coordination numbers.) (c) Energy profiles of possible reaction pathways.

In response to your constructive feedback, we have incorporated the results of the XPS, XAES analysis, and the theoretical simulations into our revised manuscript, please refer to the highlighted revision made on page 7-10, and the modified Figs. 2e, 2f, 3d, 3e, 3f, and Supplementary Fig. 8. We have also supplemented the provided references (DOI: 10.1149/2.0461904jes) to ensure a thorough discussion (refer to the newly added Ref. 46). We believe these additional data and discussion provide substantially support for our proposed charging mechanism, thereby enhancing the understanding of the reaction processes involved. We appreciate your valuable suggestions, which have allowed us to improve the clarity and accuracy of our

manuscript.

3. Line 171 Two different XPS spectra should be presented—after charging to 0.3V (formation of the intermediate phase), and after Cu deposition. This will give appropriate information about the changes in the electrode surface as a function of the applied potential.

Response:

We appreciate your insightful comment suggesting the inclusion of two distinct XPS spectra to better illustrate the surface changes of the electrode in response to the applied potentials. We concur with your proposal that presenting an XPS spectrum after charging to 0.3 V (correlating to the formation of the intermediate phase) and another subsequent to copper deposition would provide valuable insights.

In response to your suggestion, we have performed additional XPS and XAES measurements on the electrode surface at these specific voltage states. The analysis of the spectra after copper deposition discloses the presence of copper in 0, +1, and +2 valence states (Fig. R4a, refer to *Angew. Chem. Int. ed.*, 2022, 134, e202212191, *J. Am. Chem. Soc.*, 2013, 135, 14032-14035, and *ACS Sustain. Chem. Eng.*, 2022, 10, 15958-15967). The 0 valence is indicative of the deposited copper, while the +1 and +2 valence states are associated with the residual cuprous intermediate phase and CuCl_2 from the electrolyte, respectively.

Fig. R4. Electrode surface valence changes during the electrochemical processes. (a) XPS spectra and (b) XAES spectra corresponding to charging to 0.3 V and after Cu deposition.

Furthermore, compared to the spectrum correlating to the state after Cu deposition, the copper LMM spectrum after charging to 0.3 V reveals the disappearance of 0-valent copper and the predominant presence of +1 and +2 valence states (Fig. R4b, refer to *Adv. Mater.*, 2022, 34, 2205229 and *ACS Sustain. Chem. Eng.*, 2022, 10, 15958-15967). This observation supports our proposed charging mechanism, indicating the oxidation of copper and the formation of the cuprous intermediate phase.

Your valuable suggestion has indeed refined the specificity and comprehensiveness of our investigation. In our revised manuscript, these XPS and XAES spectra have been included (refer to the highlighted revision made on page 9 and the modified Figs. 3d, 3e, and the newly added Supplementary Fig. 10), offering readers a clearer understanding of the surface changes during the electrochemical processes.

4. Line 195 To which potential the b values are correlated? Please show the full CV response from which the contribution ratio (Fig. 3d was calculated)

Response:

Thank you for your valuable feedback. The b values are correlated to the oxidation

peaks at around -0.2 and 0.4 V (*vs.* SCE) for chlorine coordination, and around 0.15 V (*vs.* SCE) for water coordination, as labelled by Roman numerals (b_I, b_{II}, b_{III} for peak I, II, and III, respectively).

We apologize for not including the complete CV response in the manuscript. The contributions shown in Fig. 3d were calculated from the CV curves of the Cu–Cl and Cu–H₂O electrolytes (shown in Fig. R5 and R6). We have incorporated these CV curves in the revised manuscript, please refer to the newly added Supplementary Figs. 13 and 14.

Fig. R5 The CV curves for Cu–Cl electrolyte at various scan rates (the shaded areas show the calculated capacitive contributions). (a) 1 mV s^{-1} . (b) 2 mV s^{-1} . (c) 4 mV s^{-1} . (d) 6 mV s^{-1} . (e) 8 mV s^{-1} . (f) 10 mV s^{-1} .

Fig. R6 The CV curves for Cu-H₂O electrolyte at various scan rates (the shaded areas show the calculated capacitive contributions). (a) 1 mV s⁻¹. (b) 2 mV s⁻¹. (c) 4 mV s⁻¹. (d) 6 mV s⁻¹. (e) 8 mV s⁻¹. (f) 10 mV s⁻¹.

5. A general note While the performance of the full cell is presently nice, the author should also indicate what are the practical limitations of this system. Clearly, as Cl₂ is a volatile gas, such a system can not be operated at very slow charging rates which obviously results in the generation of a significant amount of hazardous Cl₂ gases. In addition, I expect that the self-discharge of the system will be dramatically fast (as Cl₂ will not remain on the surface for a long time). Nevertheless, I think that from a scientific point of view, this manuscript is very interesting. Yet as scientists, we have to point out what are the limitations and the challenges of this system. This in my view will not reduce the importance and effectiveness of the work, but oppositely will be highly appreciated by readers and will help to further develop aqueous batteries for energy storage applications.

Response:

Thank you for your insightful comments and highlighting the practical limitations of the proposed system. We agree that it is vital to acknowledge the challenges and limitations of the Cu-Cl₂ battery system for its potential applications.

As you pointed out, one of the primary concerns associated with this system pertains to the involvement of Cl_2 gas, which poses potential safety risks. Given that Cl_2 is a volatile and hazardous gas, its generation during the battery operation, especially at slow charging rates, could necessitate the installation of safety measures such as ventilation or containment systems to prevent inadvertent release. It is crucial to note that these safety requirements, while essential, might complicate the design and operational logistics of the battery system.

Furthermore, the volatility of Cl_2 also raises concerns regarding the self-discharge of the Cu- Cl_2 battery system, potentially reducing the efficiency and performance of the battery. This phenomenon has not been thoroughly explored within this study, and we see this as an essential area for future research to fully understand and mitigate this issue.

Despite these challenges, the study's findings suggest the feasibility of Cl^-/Cl_2 as a cathode reaction, opening new possibilities for the development of energy storage solutions. The incorporation of 0.5 M HCl into the Cu-Cl electrolyte led to improved stability, achieving 88% capacity retention after 5000 cycles. It is evident from this research that a balance between maximizing the potential advantages and minimizing the possible risks associated with the Cu- Cl_2 battery system can be achieved, leading to further development and optimization.

Therefore, our study provides an important starting point in the investigation of Cu- Cl_2 batteries, and although several practical challenges need to be overcome, the potential benefits and advancements in the field of energy storage are significant. These challenges also present opportunities for further investigation and continuous improvement in the design and performance of such batteries. We remain committed to advancing this field of research and exploring innovative solutions to address these challenges.

We are appreciative of your supportive remarks on the scientific value of our work, and we have supplemented the discussion on these challenges and limitations in the revised manuscript, please refer to the highlighted revision made on page 16.

Minor comments

6. Line 46 – Please add also Mn ion battery which also starts to attract attention. The following references are recommended (10.1021/acseenergylett.2c02242, 10.1002/ange.202200809)

Response:

Thank you for the insightful suggestion. We agree that this research area is indeed noteworthy and attracting attention in the field.

In response to your comment, we have incorporated Mn ion battery in the introduction section. We have also supplemented the provided references (DOI: 10.1021/acseenergylett.2c02242, DOI: 10.1002/ange.202200809) to ensure a thorough and up-to-date discussion. Please refer to the highlighted revision made on page 3, and the newly added references Ref. 16 and Ref. 17.

7. Line 96 As far as I know, NaF can not reach 4 M in an aqueous medium. What is the actual concentration of the saturated NaF solution?

Response:

Thank you for your keen observation and comment regarding the concentration of the NaF solution. We realize, upon your astute remark and further investigation, that our initial statement indicating a 4 M concentration of NaF in an aqueous medium is inaccurate. The actual solubility of NaF in water at room temperature is roughly 4.2 g per 100 mL at 25 °C, which corresponds to a saturated solution concentration of approximately 1 M.

We sincerely apologize for the confusion our initial statement may have caused, and we are grateful for your vigilance in bringing this matter to our attention. We have promptly amended this error in our revised manuscript, ensuring the accurate information is presented. Please refer to the highlighted revision made on page 4.

8. Line 159 It seems that the reference number ‘jumped’ into the oxidation number

notation of the Cu.

Response:

Thanks for the careful review. We regret for this oversight. Accordingly, we have corrected the numbering for all the references, please check the revised manuscript.

To Reviewer #2

Overall comment:

The work suggested through manipulating coordination chemistry of Cu–Cl₂ for a high voltage aqueous battery is conceptually good potential of interest. Despite the authors have shown valuable results on electrochemical performance/properties, there is a huge gap in explaining the coordination chemistry of Cu–Cl, which is important as mentioned in the title of the manuscript, resulting lack of evidence for proving their origin of formations. The reviewer cannot recommend this manuscript for publication in Nature Communications. Few following comments can help the authors.

Response:

We are grateful for your insightful comments regarding our manuscript and your recognition of the novelty of our work. We acknowledge the importance of providing a comprehensive explanation for the coordination chemistry of Cu–Cl, as stated in our manuscript title.

We understand your concern about the current lack of thorough evidence supporting our findings. We concur that addressing this issue is pivotal for our work to meet the publication standards of *Nature Communications*. Therefore, we are committed to providing additional substantial evidence to substantiate our claims surrounding the coordination chemistry of Cu–Cl, aiming to enhance the clarity and completeness of our manuscript through these amendments.

1. There is an empty space to explain the Cu–Cl complex coordination, questioning basic chemistry behind solvation of salts in water. What is the coordination chemistry of CuCl₂. 2H₂O + 4M NaCl in aqueous medium? Wondering that it is not a very high concentrated solution like water-in-salt solution to completely repel all water molecules surrounded by Cu for coordination, then how the coordination has derived in absence of H₂O molecules in a large pool of aqueous solution?

Response:

We appreciate your valuable comments and the concerns raised regarding the explanation of Cu–Cl coordination chemistry. We understand the need for clarity in depicting the fundamental chemistry behind the solvation structure, particularly focusing on the coordination chemistry of $\text{CuCl}_2 + 4 \text{ M NaCl}$ in aqueous medium.

In an aqueous medium, CuCl_2 dissociates into Cu^{2+} and Cl^- ions. Cu^{2+} , as a Lewis acid, is capable of coordinating with both water molecules and chloride ions. The coordination behaviour of Cu^{2+} in the presence of chloride ions primarily depends on the concentration of these ions. At low chloride ion concentrations, Cu^{2+} predominantly coordinates with water molecules, forming $[\text{Cu}(\text{H}_2\text{O})_6]^{2+}$ complexes through hydrolysis (*Nat. Commun.*, 2023, 14, 2349). As chloride concentration increases, these ions compete with water molecules for Cu^{2+} coordination, leading to the formation of Cu–Cl complexes.

To thoroughly investigate the coordination chemistry of Cu^{2+} , we conducted surface-enhanced Raman scattering (SERS) on solutions with varying chloride ion concentrations. Our findings revealed a shift in Cu^{2+} coordination as chloride ion concentration increased, indicated by a change in the solution color from blue to green (Fig. R7a). Additionally, the Cu–Cl stretching Raman signal at approximately 275 cm^{-1} intensified (Fig. R7b), validating the coordination of Cu^{2+} with chloride ions (*Sci. rep.*, 2015, 5, 13759; *Chem. Commun.*, 2022, 58, 10076-10079).

Fig. R7 The Cu–Cl complex coordination. (a) Optical photos of CuCl_2 in different electrolytes. (b) The SERS spectra of CuCl_2 in various NaCl solutions.

Notably, Cu–Cl coordination does not necessitate exceedingly high chloride ion concentrations and generally occurs at concentrations above 1 M. This is due to the higher negative charge and larger size of chloride ions, which allow for the formation of stronger bonds with positively charged copper ions compared to water molecules.

We acknowledge that in an aqueous medium, Cu^{2+} is indeed surrounded by numerous water molecules. However, the coordination of Cu^{2+} with its ligands is a dynamic process, characterized by rapid ligand exchange, such as between water molecules and chloride ions. Importantly, the coordination of copper with chlorine does not entirely exclude water molecules from the coordination sphere, resulting in a $[\text{Cu}(\text{H}_2\text{O})_{6-x}\text{Cl}_x]^{2-x}$ complex that is co-coordinated with water. This behaviour is a common occurrence in coordination chemistry, where a metal ion can simultaneously coordinate with multiple ligands.

We sincerely apologize for any confusion our previous statements may have caused and have taken steps to clarify these points in our revised manuscript. Please refer to the highlighted revision made on page 7 and the modified Supplementary Fig. 7.

2. What is the structure and coordination numbers of each intermediate products formed during reaction? For example, what would be the structure/coordination number of $[\text{CuCl}_3]^{2-}$ as an intermediate species formed during reduction? There is no structural evidence. Explain in detail. Why not the coordination would be like $[\text{CuCl}_2(\text{H}_2\text{O})]^-$, in which Cu stands at +1 oxidation state.

Response:

Thanks for your valuable and helpful comments. We sincerely apologize for the lack of clarity and incomplete discussion regarding the structure and coordination numbers of intermediate products formed during the reaction. To address this concern, we conducted molecular dynamics (MD) simulations to analyze the solvation structure and coordination number of various electrolytes. Our findings indicate that in the Cu–H₂O electrolyte, copper coordinates with six water molecules, whereas in the

Cu–Cl electrolyte, the copper-water-chloride complex contains 1.8 chlorine and 4.2 water molecules (Fig. R8). These results suggest that in the CuCl₂–NaCl₂ solution with 4 M NaCl, a chloride-rich hydration sheath formed around copper rather than the complete hydration of copper ions.

Fig. R8 Coordination numbers of copper ions in the Cu–Cl and Cu–H₂O electrolytes. (a) RDF of the Cu–H₂O electrolyte. (b) RDF of the Cu–Cl electrolyte. The solid lines are the radial distribution functions, and the dotted lines represent the coordination numbers.

Furthermore, our Raman spectroscopy results reveal a broad peak observed at 275 cm⁻¹ (Fig. R7b), indicating the presence of various coordination complexes such as CuCl⁺, CuCl₂⁰, CuCl₃⁻, and CuCl₄²⁻ (*Sci. Rep.*, 2015, 5, 13759; *Chem. Commun.*, 2022, 58, 10076-10079). Therefore, it is more accurate to represent the copper chloride complexes as [CuCl_x]^{2-x}. Accordingly, the corresponding electrochemical process can be described as follows:

Meanwhile, the presence of a peak at 270 nm in the UV-Vis spectra during the charging and discharging process provides evidence that the main coordination form of the intermediate product is CuCl₂⁻/CuCl⁰ (*Geochim. Cosmochim. Acta*, 2007, 71, 4920-4941; *Geochim. Cosmochim. Acta*, 2002, 66, 3615-3633), supporting the existence of

multiple forms of intermediate products. Hence, the expression of CuCl_x^{1-x} is a more appropriate representation. This approach enables a clear representation of our core issue, which is the occurrence of two redox reactions of the copper anode under chlorine coordination: $\text{Cu}^+/\text{Cu}^{2+}$ and Cu^+/Cu^0 , respectively.

We have thoroughly revised our manuscript to incorporate these new findings and provided a more comprehensive and detailed understanding of copper solvation structure and reaction mechanisms. Please refer to the highlighted revision made on page 7, 8, 10, and the modified Figs. 2e, 2f, and Supplementary Fig. 8. We deeply appreciate your valuable feedback.

3. To establish the electrochemical mechanisms further, analyzing the residual electrolytes and the carbon electrodes (both carbon felt/cloth and KJB) available during the electrochemical reactions for both cases, are strongly required. There is a need of strong evidence to confirm the $\text{Cu}^{2+}/\text{Cu}^+$ and Cu^+/Cu^0 through valuable spectroscopy techniques to support the UV-Vis analysis. Find a suitable analysis that helpful to understand both oxidation states and coordination chemistries of Cu in both electrolytes (CuCl_2 and CuSO_4). Hint. Resonance spectroscopy.

Response:

Thanks for your valuable and insightful comments. We sincerely appreciate your suggestions regarding the need for additional analysis of the residual electrolytes and carbon electrodes to further establish the electrochemical mechanisms. To address this concern, we performed ex-situ XPS and XAES on the carbon cloth coated with activated carbon, known for its strong adsorption capacity. These spectroscopic techniques allowed us to probe the valence states of copper at different voltage states, providing valuable insights into the electrochemical processes.

The analysis of the XPS spectra reveal distinct peaks corresponding to different Cu species. As shown in Fig. R9, the peaks observed at approximately 935.2 and 955 eV, assigned to Cu $2p_{3/2}$ and Cu $2p_{1/2}$, respectively, indicated the presence of Cu^{2+} species. In contrast, the peaks at 932.3 and 952 eV are attributed to Cu^+ and/or metallic

Cu⁰ species. This spectroscopic evidence confirms the existence of Cu²⁺/Cu⁺ and Cu⁺/Cu⁰ redox couples during the electrochemical reactions. Further characterization using XAES allowed us to distinguish the chemical states of Cu⁰ and Cu⁺. The peak fitting analysis of Cu LMM spectra demonstrates that the peak at around 918.5 eV corresponds to metallic Cu⁰, the peak at 915.5 eV is attributed to Cu⁺, and the peak at around 917 eV is associated with Cu²⁺. These findings provide additional confirmation of the different oxidation states of Cu present during the electrochemical processes.

Fig. R9. Copper ion electrochemical behaviours under chlorine coordination. (a) CV curve obtained at a scan rate of 1 mV s⁻¹ for Cu-Cl electrolyte. (b) *Ex situ* XPS spectra corresponding to different potentials marked in (a). (c) *Ex situ* XAES spectra corresponding to different potentials marked in (a).

Based on the comprehensive valence analysis, we can establish the electrochemical redox process of Cu. In the initial state (I), Cu predominantly exists in the +2 oxidation state. During the CV negative scan process (I→III), the Cu²⁺ species undergo partial reduction to Cu⁺/Cu⁰. Subsequently, in the positive scan process (III→V), the Cu⁰ species is oxidized back to Cu⁺ and further to Cu²⁺. These results align well with the findings from our UV-Vis analysis, strengthening the consistency and reliability of our proposed electrochemical mechanisms. The spectroscopic analysis of residual electrolytes and carbon electrodes, supported by XPS and XAES techniques, has provided compelling evidence and a more comprehensive understanding of the electrochemical processes involved. We are committed to incorporating these findings into our revised manuscript, ensuring the thorough presentation of the electrochemical mechanisms with scientific rigor and accuracy.

Furthermore, in addition to the above analyses, we have conducted additional investigations to analyze the coordination environment of the two electrolytes using SERS and nuclear magnetic resonance (NMR). These techniques provided valuable insights into the coordination chemistry of copper ions in the Cu–H₂O and Cu–Cl electrolytes.

As shown in Fig. R10a, Raman spectra revealed distinct coordination patterns between copper ions and the respective electrolytes. In the Cu–H₂O electrolyte, copper ions were primarily observed to coordinate with water molecules. However, in the Cu–Cl electrolyte, the coordination between copper ions and water was significantly weakened, and the predominant coordination was observed with chlorine. This observation was supported by the NMR spectra shown in Fig. R10b, where the peak corresponding to the coordination water in the Cu–Cl electrolyte exhibited a shift compared to the Cu–H₂O electrolyte, indicating a decrease in coordination water and the partial substitution by chloride ions (*Angew. Chem. Int. Ed.*, 2021, 133, 18395-18403; *Angew. Chem. Int. Ed.*, 2023, 135, e202218454).

Fig. R10. The coordination environment of copper ions in different electrolytes. (a) Raman spectra and (b) ^1H NMR spectra in Cu-Cl and Cu-H₂O electrolytes.

To gain a deeper understanding of the coordination chemistry, we conducted molecular dynamics simulations (shown in Fig. R8 in the response to Comment No.2). The simulated results revealed that in the Cu-H₂O electrolyte, copper ions coordinated with six water molecules. While in the Cu-Cl electrolyte, the copper-water-chloride complex consisted of approximately 1.8 chlorine ions and 4.2 water molecules. This finding suggests that in the CuCl₂-NaCl₂ solution with 4 M NaCl, a chloride-rich hydration sheath of copper was formed rather than complete hydration of the copper ions.

We deeply appreciate your suggestion to conduct more detailed analyses. By utilizing SERS, NMR, and molecular dynamics simulations, our findings provide a profound understanding of the coordination chemistry of copper ions in the respective electrolytes, offering robust evidence for the presence of Cu²⁺/Cu⁺ and Cu⁺/Cu⁰ species during the electrochemical reactions. These results have been incorporated into our revised manuscript to enhance the scientific rigor and comprehensiveness of our study. Please refer to the highlighted revision made on page 7-10 and the modified Figs. 2b, 2c, 2e, 2f, 3d, and 3e.

4. How UV-Vis analysis have been measured? With or without dilution? In such a high concentration of electrolyte, dilution is commonly employed to avoid peak amplitude

saturation and in doing so, the coordination chemistry will be changed. Explain briefly.

Response:

Thank you for your valuable comments. The UV-Vis analysis was conducted to measure the absorption spectra of Cu^+ and Cu^{2+} ions in solution, which allows for their differentiation based on their different electronic configurations. To avoid peak amplitude saturation and ensure accurate measurements, we diluted the concentration of copper ions to approximately 10 mM while maintaining a high concentration of chloride ions at 4 M. By diluting the copper ion concentration, we were able to obtain absorption spectra that provided reliable information about the coordination chemistry.

As the reviewer has pointed out, dilution is commonly employed in UV-Vis analysis of high-concentration electrolytes to mitigate the saturation of peak amplitudes. However, it is important to consider that dilution can potentially alter the coordination chemistry. In our study, 4 M NaCl was adopted as the blank sample to deduct background absorption. Although the high-concentration sodium chloride did have some absorption in the UV region, it was relatively weak and below 190 nm. Thus, the concentration of NaCl did not significantly affect the results.

We primarily focused on the coordination behavior of Cu in chloride forms, which consists of solely mononuclear complexes. The distribution of these complexes only depends on the concentration of free chloride ions rather than the total Cu concentration (*Anal. Chem.*, 2013, 85, 7696-7703). Therefore, to stabilize the concentration of free chloride ions and minimize any changes in copper chloride coordination, we maintain a high concentration of chloride ions at 4 M. Electrochemical measurements, as indicated by the CV curves, confirmed that the electrochemical behavior of the diluted solution remained unchanged. Hence, in combination with other characterization data, we have acquired reliable results about the Cu coordination chemistry, and the $\text{Cu}^+/\text{Cu}^{2+}$ redox reaction is accordingly proposed.

In addition, we have updated our description of copper chloride coordination in the revised manuscript to reflect the mixed coordination behaviour using $[\text{CuCl}_x^{2-x}]$ as the notation. This notation more accurately represents the two-step redox process of the

copper anode, as specified in our response to Comment No. 2.

Thank you for bringing these points to our attention, and we have incorporated these clarifications into our revised manuscript to ensure the accuracy and comprehensibility of our findings. Please refer to the highlighted revision made on page 8 and 10.

5. How to control the further dissociation of HClO into hypochlorite (OCl^-), resulting in formation of H^+ in presence of strong 0.5 M HCl added electrolyte? What would be a pH value? The above said dissociation reaction could be spontaneous reaction ($\text{Cl}_2 + 2 \text{H}_2\text{O} \rightarrow \text{HOCl} \rightarrow \text{H}^+ + \text{OCl}^-$)!

Response:

Thanks for your valuable comments. To control the further dissociation of HClO into hypochlorite and H^+ , maintaining a low pH is key. In our study, 0.5 M HCl was added in the electrolyte, as HCl is a strong acid, the dissociated H^+ helps shift the reaction equilibrium towards HClO ($\text{HClO} \leftrightarrow \text{H}^+ + \text{ClO}^-$), thereby minimizing the formation of hypochlorite ions (ClO^-). The pH value of a 0.5 M HCl solution is approximately 1.3.

Additionally, a strong acid solution can effectively inhibit the reaction of chlorine with water ($\text{Cl}_2 + \text{H}_2\text{O} \leftrightarrow \text{HCl} + \text{HOCl}$), thus reducing the production of HClO from the chlorine source. This inhibition was confirmed in the UV test, where ClO^- was not detected, and is believed to contribute to the improved stability of the system.

Thank you for bringing up these points, and we have incorporated these explanations into our revised manuscript. Please refer to the highlighted revision made on page 14.

6. Why the cathode reaction of Cl^-/Cl^0 has been shown in terms of gram current density (mA g^{-1}) rather than areal density (mA cm^{-2}) as given for anode reaction? Similarly, the fully developed cells too represent by mA g^{-1} .

Response:

Thank you for your insightful query. We appreciate your attention to detail, and we would like to justify the rationale behind our choice of current density unit in our study. The decision to use gram current density (mA g^{-1}) rather than areal current density (mA cm^{-2}) for the cathode reaction and the full cell was based on several considerations.

Firstly, the use of gram current density is prevalent in the field when characterizing the electrochemical performance of cathode materials (*Nature*, 2021, 596, 525-530; *Nature*, 2019, 569, 245-250; *Nat. Commun.*, 2023, 14, 1856). This unit allows for direct comparison with results reported in literatures, where cathode reactions are often expressed in terms of the mass of the active material in the electrode. By employing gram current density, we have ensured compatibility with existing studies and facilitated meaningful comparisons.

Secondly, the cathode reaction in our system involves the redox process of chloride ions (Cl^-/Cl^0), where the mass of the active material in the cathode plays a vital role. Thus, expressing the current density in terms of gram (mA g^{-1}) provides a comprehensive understanding of the electrochemical performance of the cathode materials and accurately represents the underlying processes.

Conversely, the anode in our study solely consists of carbon felt, which does not contain any electrochemically active substance. The oxidation-reduction reactions occurring at the anode primarily involve copper ions from the electrolyte. Consequently, expressing the anode reaction in terms of areal density (mA cm^{-2}) was deemed appropriate. Areal density allows us to effectively evaluate the electrochemical performance per unit surface area, which is crucial for understanding the utilization and efficiency of the anode material (*Nat. Commun.*, 2022, 13, 7922; *Nat. Commun.*, 2023, 14, 2925).

By employing distinct density units for cathode and anode, we have been able to accurately reflect the specific electrochemical processes occurring at each electrode and to provide a comprehensive understanding of the overall system performance. We recognize the importance of the choice of current density unit and appreciate your

valuable feedback.

Additional comments:

7. Typos errors: line 159 Cu²⁺25. May the reference number interfere?

Response:

Thank you for your thorough review. We apologize for the oversight in the format error of the quotations. We have now corrected the numbering for all the references, please check the revised manuscript.

8. Fig. 3h, y-axis capacity retention (%) might be wrong.

Response:

We appreciate your attentive and detailed review. Upon careful examination, we recognize that the y-axis labeling in Figure 3h, denoted as "capacity retention (%)", might indeed be misleading due to excessive color usage. Your constructive feedback prompted us to make necessary modifications to enhance the clarity and precise representation of our data. The amended Fig. 3h (now as Fig. 4h in the revised manuscript) offers a more comprehensible and straightforward visualization of the capacity retention rate.

To Reviewer #3

Overall comment:

This manuscript reported a coordination strategy for reducing the intrinsic anode redox potential in aqueous copper-based batteries to improve the operating voltage. The stabilized Cu^+ ions by chlorine coordination present a redox potential approximately 0.3 V lower than that for $\text{Cu}^{2+}/\text{Cu}^0$. The Cl^-/Cl^0 redox reaction was further employed as the cathode to achieve a novel aqueous Cu-Cl₂ battery with a high discharge voltage of 1.3 V and good stability. Overall, it is an interesting work and may provide a new insight into how the coordination environment influences the anode potential and the electrochemical performance for aqueous batteries. Thus, I recommend its publication on Nature Communications after addressing the following minor issues.

Response:

Thank you for your thoughtful review and for recommending our manuscript for publication. We appreciate your recognition of the significance of our research in understanding the impact of the coordination environment on the anode potential and the overall electrochemical performance of aqueous batteries. In response to your suggestions, we have thoroughly addressed each point and incorporated additional analysis of the underlying mechanism into our manuscript. We believe these modifications have strengthened our research presentation and thank you once again for your valuable feedback.

1. Why did the authors select the 4 M NaCl and 0.05 M CuCl₂ as the electrolyte, which was not mentioned in the manuscript? How about its performance with other ratios? The authors should explain their choice on electrolyte more profoundly.

Response:

Thanks for your insightful query concerning the rationale behind our selection of the 4 M NaCl and 0.05 M CuCl₂ as the electrolyte. We acknowledge the oversight in

our original manuscript and have now elaborated on this in the revised version.

Our choice of electrolyte was guided by several key factors, including specific capacity, Coulombic efficiency, and the reversibility of the redox reactions. We conducted a series of tests using various NaCl concentrations (Fig. R11a-c) and observed a positive correlation between NaCl concentration and Coulombic efficiency. This efficiency reached a maximum at 4 M NaCl, beyond which no further improvement was noted. Additionally, the areal capacity remained consistent across varying concentrations. Therefore, we selected 4 M NaCl as the fixed concentration for our experiments.

Fig. R11 The influences of electrolytes. (a) CV curves, (b) GCD curves, and (c) the Coulombic efficiency with varying NaCl concentrations. (d) CV curves, (b) GCD curves, and (c) the Coulombic efficiency with varying CuCl_2 concentrations.

In parallel, we also studied the impact of differing CuCl_2 concentrations on electrode performance (Fig. R11d-f). At elevated CuCl_2 concentrations, we observed an increase in areal capacity; however, the Coulombic efficiency was adversely affected, signifying poor reversibility. Conversely, lower CuCl_2 concentrations resulted in high Coulombic efficiency but at the expense of areal capacity. Balancing these considerations, we chose 0.05 M CuCl_2 as the optimal concentration for our

experiments.

Overall, the decision to utilize a 4 M NaCl and 0.05 M CuCl₂ electrolyte was informed by its resultant high areal capacity, Coulombic efficiency, and good reversibility of the redox reactions. We have now expanded upon this explanation in the revised manuscript to ensure our choice of electrolyte is fully justified and transparent. Please refer to the highlighted revision made on page 6 and the modified Supplementary Figs. 3 and 4.

2. The authors used carbon cloth as the anode and actually constructed an anode-free Cu-Cl₂ battery. How does it perform when metallic copper is used as the anode? What are the advantages of carbon cloth in comparison?

Response:

Thanks for your insightful comments. In our study, we opted to use carbon cloth as the anode material due to its superior electrochemical performance, including high specific capacity and stability. Through our investigations, we discovered notable advantages using carbon cloth over metallic copper.

Under chlorine coordination, copper ions undergo a two-step reaction: $[\text{CuCl}_x]^{2-x} + e^- \leftrightarrow [\text{CuCl}_x]^{1-x}$ (I) and $[\text{CuCl}_x]^{1-x} + e^- \leftrightarrow \text{Cu}^0 + x\text{Cl}^-$ (II). When metallic copper is used as the anode, the two-step reaction of copper ions is hindered, as the copper must be fully engaged in reaction (II) before proceeding with reaction (I). Additionally, the process would be further complicated by the excessive precipitation and detachment of copper on the copper anode surface, leading to low Coulombic efficiency, as shown in Fig. R12a.

Fig. R12 Comparison between metallic Cu anode and carbon cloth anode. (a) GCD curve of Cu anode. (b) GCD curve of carbon cloth anode.

In contrast, the use of carbon cloth as the anode material enabled the reversal of the two-step reaction of copper ions under chlorine coordination, resulting in high Coulombic efficiency (Fig. R12b). Furthermore, carbon cloth demonstrates high electrical conductivity, good stability, and promotes efficient mass transport and electrochemical reactions, thereby enhancing the overall performance of the battery. Carbon cloth also exhibits good resistance to both passivation and dissolution, leading to better battery performance and stability compared to metallic copper.

In conclusion, our study shows that carbon cloth is a suitable and superior choice for anode material in the Cu-Cl₂ battery system. The use of carbon cloth in an anode-free Cu-Cl₂ battery improved the electrochemical performance, marked by high specific capacity and stability, offering significant advantages over the conventional metallic copper anode.

3. Fig. 3g, the copper anode under chlorine coordination shows higher rate capability than that under water coordination, which means the faster electrochemical kinetics for chlorine coordination. What is the specific reason? Please clarify this point.

Response:

Thank you for your valuable comments. Your query prompted us to delve deeper into the specifics of the charge storage kinetics under these two distinct coordination

environments. Our investigation revealed that the b values calculated for the two redox pairs (I and II) under chlorine coordination are 0.46 (b_I) and 0.77 (b_{II}), respectively, in contrast to a b value of 0.30 (b_{III}) for the redox pair III under water coordination. These values suggest that the electrochemical reactions of redox I and III followed slow diffusion-controlled processes, while redox II proceeded through a faster pseudocapacitive-controlled process.

Further insights from XRD analysis reveal that redox II involved a liquid-liquid reaction ($[CuCl_x]^{2-x} + e^- \leftrightarrow [CuCl_x]^{1-x}$), compared to the solid-liquid reactions ($[CuCl_x]^{1-x} + e^- \leftrightarrow Cu^{0+x}Cl^-$ and $Cu^{2+} + 2e^- \leftrightarrow Cu^0$) in redox I and III which typically show slower kinetics. Thus, the liquid-liquid reaction in redox II allowed for more efficient charge transfer, as it can bypass the solid-state interface.

In addition, we noted that the capacitive contributions under chlorine coordination are higher than those under water coordination. This observation points to faster reaction kinetics under chlorine coordination, which we attribute to the stronger interaction between copper and chlorine ions that promotes more efficient charge transfer.

Therefore, the superior rate capability of the copper anode under chlorine coordination can be ascribed to the synergistic effect of a rapid liquid-liquid reaction and increased capacitive contributions. This analysis has been incorporated into the revised manuscript to provide a clearer and more nuanced understanding of the faster kinetics observed under chlorine coordination. Please refer to the highlighted revision made on page 11.

4. The authors used carbon cloth as a cathodic current collector. Can it adsorb molecular chlorine to contribute to the capacity? What are the criteria for selecting the current collector?

Response:

Thanks for your valuable comments. In response to your question regarding the potential adsorption of molecular chlorine by the carbon cloth current collector, we

have conducted additional investigations. As shown in Fig. R13, our findings reveal that pure carbon cloth had a negligible contribution to the battery's capacity and it exhibited significant redox polarization when exposed to chlorine. However, when equipped with Ketjen black (KJB), the carbon cloth exhibited a notable decrease in polarization and an increase in reversible capacity, suggesting a pronounced adsorption effect of KJB on chlorine.

Fig. R13 Contribution of carbon cloth to the capacity. (a) GCD curve of pure carbon cloth. (b) GCD curve of carbon cloth + KJB.

Selecting the optimal current collector is indeed a critical aspect in battery system design. A current collector essentially serves as an electronic bridge to the external circuit, significantly influencing the electrochemical performance of the battery, including its capacity, cycling stability, and rate capability. Our selection criteria for the cathodic current collector encompassed several key factors, including electrical conductivity, chemical stability, mechanical strength, and surface area.

The electrical conductivity of current collectors is vital as it minimizes energy loss during the charge and discharge cycles. Equally important is chemical stability, which ensures resistance against corrosion and degradation during electrochemical reactions. Mechanical strength is also required to uphold the battery's structural integrity during cycling, and a high surface area is beneficial for boosting reaction kinetics and mass transport of the active materials.

In light of these considerations, we selected carbon cloth as the cathodic current

collector in our Cu-Cl₂ battery system. Its established application as a current collector in various battery systems and its cost-effectiveness further substantiated our choice for this study. We have incorporated the detailed explanation into the "Methods" section. Please refer to the highlighted revision made on page 19 and the modified Supplementary Fig. 19.

5. Besides the hydrolysis of molecular chlorine ($\text{Cl}_2 + \text{H}_2\text{O} \leftrightarrow \text{HCl} + \text{HClO}$), the competing oxygen evolution reaction (OER) should also receive attention. The authors need to evaluate the oxygen evolution reaction (OER) process after the incorporation of HCl. In this case, the suppression of side reactions can be well studied.

Response:

We appreciate your insightful comment regarding the necessity of investigating the OER process in the Cu-Cl₂ battery system. Indeed, the suppression of OER is a critical aspect to improve the overall electrochemical performance of the battery system. In response, we have conducted additional experiments to evaluate the impact of HCl on OER, which will shed light on the battery system's stability and efficiency.

To accurately track gas production, we employed an *in-situ* gas chromatography system, allowing real-time monitoring oxygen production. The results in Fig. R14 reveal that without the addition of HCl, the oxygen production rate was relatively high (over 0.1 ml h⁻¹), with a steady increase over time. However, upon the addition of 0.5M HCl, the oxygen production rate significantly decreased (down to approximately 0.03 ml h⁻¹), indicating the effective suppression of side reactions. This observation strongly evidences that the incorporation of HCl can effectively deter the OER process, thereby improving the battery system's overall performance.

Fig. R14 *In-situ* monitoring of oxygen production on an *in-situ* gas chromatography system.

It is noteworthy that distinguishing between OER and the chlorine evolution reaction (CIER) solely through electrochemical testing is challenging due to their competitive nature. Therefore, the implementation of an *in-situ* gas chromatography system in our experiment provides a reliable method to evaluate the inhibitory effect of HCl on OER and its consequential impact on the overall electrochemical performance of the battery system.

We have updated the manuscript with these findings to clarify the role of HCl in mitigating side reactions. Please refer to the highlighted revision made on page 14 and the modified Supplementary Fig. 16.

6. The Cu-Cl₂ full cell delivered high energy and power densities of 141.2 Wh kg⁻¹ and 4236 W kg⁻¹, respectively. How does it calculate? Is it based on the mass of both cathode and anode active materials?

Response:

Thank you for your constructive comment. The energy and power densities for the Cu-Cl₂ full cell reported in our study were solely determined based on the mass of the cathode active materials. This is due to the fact that the anode design does not

incorporate any active substance and consists only of a carbon cloth collector.

To calculate the energy density (E), we used the equation $E = \int_0^Q V(q) dq$, where Q is the specific capacity of the cell, V is the voltage of the cell, q denotes the state of discharge, and $V(q)$ is the voltage of the cell corresponding to the state of discharge q . The power density was evaluated using the formula $P = E/t$, with t representing the working time. By factoring in these metrics, we ensure a comprehensive and precise measurement of our battery's performance.

We have incorporated the calculation details in the "*Methods*" section. Please refer to the highlighted revision made on page 19.

7. There are some minor issues that should be addressed, for example, Page 4, line 86, add "a" in front of high discharge voltage; Page 6, line 137, replace "was" by "were"; Page 9, line 198, replace "i(v)" by "i(V)".

Response:

Thank you for your careful review. We have revised the manuscript according to your recommendations as follows: on page 4, line 86, we have included "a" before "high discharge voltage" for grammatical accuracy; on page 6, line 137, we have corrected the verb agreement by replacing "was" with "were"; on page 9, line 198, the symbol "v" has been replaced with "V" as per your suggestion. We acknowledge the minor errors you pointed out in our manuscript and appreciate your detailed attention to the language. Please check the revised manuscript.

REVIEWERS' COMMENTS

Reviewer #1 (Remarks to the Author):

Taking into account the thorough responses, the supplementary experimental data, and the theoretical calculations, I'm pleased to say that my concerns have been sufficiently addressed.

Therefore I support the publication of this work in Nature Communications.

Reviewer #3 (Remarks to the Author):

Authors have made changes to address all reviewers' comments. I have no further comments and think this manuscript is in good shape to publish.